# From Debate to Equilibrium:
# Belief-Driven Multi-Agent LLM Reasoning via Bayesian Nash Equilibrium

Yi Xie [1 2 *]  Zhanke Zhou [2 *]  Chentao Cao [2]  Qiyu Niu [1]  Tongliang Liu [3]  Bo Han [2]

## Abstract

Multi-agent frameworks can substantially boost the reasoning power of large language models (LLMs), but they typically incur heavy computational costs and lack convergence guarantees. To overcome these challenges, we recast multi-LLM coordination as an incomplete-information game and seek a Bayesian Nash equilibrium (BNE), in which each agent optimally responds to its probabilistic beliefs about the strategies of others. We introduce *E*fficient *Co*ordination via *Nash* Equilibrium (*ECON*), a hierarchical reinforcement-learning paradigm that marries distributed reasoning with centralized final output. Under ECON, each LLM independently selects responses that maximize its expected reward, conditioned on its beliefs about co-agents, without requiring costly inter-agent exchanges. We mathematically prove that ECON attains a markedly tighter regret bound than non-equilibrium multi-agent schemes. Empirically, ECON outperforms existing multi-LLM approaches by $11.2\%$ on average across six benchmarks spanning complex reasoning and planning tasks. Further experiments demonstrate ECON's ability to flexibly incorporate additional models, confirming its scalability and paving the way toward larger, more powerful multi-LLM ensembles. The code is publicly available at: https://github.com/tmlr-group/ECON.

## 1. Introduction

Large Language Models (LLMs) have demonstrated exceptional reasoning capabilities across a wide range of tasks,

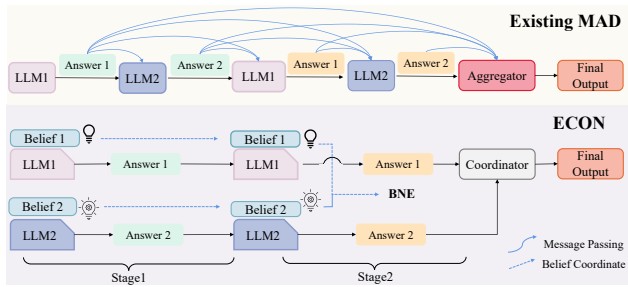

Figure 1: Comparison of multi-agent coordination approaches. Existing MAD requires explicit message passing between LLMs, incurring high communication overhead. ECON replaces it with belief-based coordination and achieves BNE—a stable state where each LLM optimizes its strategy based on beliefs about other LLMs' behaviors.

from natural language understanding and generation to complex problem-solving. Recent work has shown that organizing multiple LLMs into a *multi-agent* framework can further amplify their reasoning power by enabling collaborative in-context discussions without parameter updates (Du et al., 2024; Chan et al., 2024; Liang et al., 2023; Chen et al., 2023; Hong et al., 2023; Zhang et al., 2023b).

In particular, multi-agent debate (MAD) (Liang et al., 2023; Du et al., 2024) leverages structured argumentation among several LLM agents, where each participant critically evaluates and refines others' proposals to arrive at a more robust, consensus-driven solution. MAD has been shown to outperform single-agent methods in various reasoning scenarios, providing clearer justifications and reducing error rates.

However, existing multi-agent frameworks encounter three key obstacles that limit their practical deployment. First, extensive inter-agent communication consumes large numbers of tokens, driving up computational overhead (Du et al., 2024). Second, the sheer volume of information exchanged over multiple rounds exceeds LLMs' context-window capacity, impeding scalability (Liu et al., 2024b). Third, without well-defined coordination protocols, these systems can underperform simpler ensembling or self-consistency methods (Liang et al., 2023). Moreover, recent work has shown that the consensus threshold between agents can significantly influence performance (Smit et al., 2024). Thus,

*Equal contribution  [1]Academy for Engineering and Technology, Fudan University [2]TMLR Group, Department of Computer Science, Hong Kong Baptist University [3]Sydney AI Center, The University of Sydney.  Correspondence to: Bo Han <bhanml@comp.hkbu.edu.hk>.

*Proceedings of the 42nd International Conference on Machine Learning*, Vancouver, Canada. PMLR 267, 2025. Copyright 2025 by the author(s).

developing a principled, scalable framework for multi-agent coordination is necessary but remains an open challenge.

In this work, we introduce *Efficient Coordination via Nash Equilibrium (ECON)*, a novel framework that casts multi-LLM interaction as an incomplete-information game. As shown in Figure 1, conventional MAD approaches rely on heavy, round-by-round message exchanges among all agents, incurring substantial computational overhead. ECON, by contrast, replaces direct communication with a belief-based coordination mechanism: each LLM maintains and updates probabilistic beliefs about its peers' behaviors, rather than sending and receiving explicit messages.

Notably, we frame coordination as the pursuit of a *Bayesian Nash Equilibrium (BNE)*, where each LLM chooses its optimal strategy in light of its beliefs about other agents. Concretely, each Execution LLM uses a belief network to convert its local observations and past trajectory into a belief state, and then produces an output that maximizes its expected reward. All agents' belief states are merged by a Belief Encoder into a shared, group-level representation, which feeds into a centralized mixing network. This network refines the belief-network parameters for every agent, gradually guiding the entire ensemble toward a BNE in which every LLM converges to a stable strategy. By substituting costly token exchanges with belief-driven coordination, ECON dramatically cuts communication overhead while preserving and even enhancing multi-agent reasoning.

Theoretically, we begin by proving the existence of a BNE and then bound the performance gap to an optimal strategy via regret analysis. Specifically, ECON attains a sublinear regret of $O\left(N\sqrt{T}/1-\gamma\right)$, where $N$ is the number of agents, $T$ the number of iterations, and $\gamma$ the discount factor. By contrast, existing multi-agent frameworks lacking an equilibrium guarantee suffer linear regret $O\left(\delta_{\max}T/1-\gamma\right)$. This tighter bound demonstrates ECON's capacity to learn near-optimal strategies efficiently, with far lower token and computation costs than existing MAD methods. Consequently, ECON scales effectively, addressing the scalability limitation in prior research of multiagent (Wu et al., 2024; Yin et al., 2023; Lan et al., 2024; Yuan et al., 2024a) and graph reasoning (Zhou et al., 2023b;a; Li et al., 2024; Zhou et al., 2024b). Attributed to the local-global Nash coordination (central-coordinator-execution) design, ECON can ensemble up to nine LLMs with only moderate resource increases.

Empirically, ECON reliably drives the multi-LLM system toward a Bayesian Nash Equilibrium, fostering collaborative reasoning and stronger consensus among agents. On six diverse benchmarks—spanning complex reasoning and planning tasks—ECON outperforms single-agent baselines by an average of $10.9\%$ and existing multi-agent methods by $11.2\%$, demonstrating its superior effectiveness. Moreover, compared to a 3-round MAD protocol, ECON reduces token

usage by $21.4\%$ on average. In scalability tests, increasing the number of Execution LLMs from three to nine yields an additional $18.1\%$ performance gain, underscoring ECON's capacity to scale with only moderate resource overhead.

We summarize our key contributions as follows:

- We formalize Bayesian Nash Equilibrium for multi-agent LLM systems, establishing theoretical foundations for efficient coordination without direct communication (Sec. 2).
- We introduce the ECON framework to implement BNE via belief-based coordination and overcome scalability limits through a local–global Nash mechanism (Sec. 3).
- We empirically demonstrate that ECON outperforms both existing single-agent and multi-agent approaches, and validate its efficiency to scale to larger ensembles (Sec. 4).

## 2. Theoretical Foundation

This section formalizes BNE for multi-agent LLM systems, establishing theoretical foundations for efficient coordination without direct communication. We model the multi-agent setup as a DEC-POMDP in Sec. 2.1, prove BNE existence in Sec. 2.2, then develop a convergence analysis with performance difference lemmas and regret bounds.

### 2.1. Problem Definition

We consider a framework improving collaboration under partial observability and without fine-tuning the internal parameters of any LLM. In this setting, agents cannot observe each other's outputs directly and rely on beliefs to coordinate. We formally model this as a decentralized partially observable Markov decision process (DEC-POMDP), defined as *Markov games* $\langle \mathcal{N}, \mathcal{S}, \mathcal{A}, \mathcal{O}, \mathcal{P}, \Omega, \mathcal{R}, \gamma \rangle$. Here, $\mathcal{N} = \{1, \ldots, N\}$ denotes the set of agents; $\mathcal{S}$ represents the state space, including user queries and dialogue context; $\mathcal{A} = \mathcal{A}_1 \times \cdots \times \mathcal{A}_N$ is the joint action space, with each $\mathcal{A}_i$ defining agent $i$'s action as a prompt embedding $a_i = [T_i, p_i]$ that controls generation behavior via temperature and repetition penalty; $\mathcal{O}$ is the joint observation space; $\mathcal{P}$ and $\Omega$ define the state transition and observation functions; $\mathcal{R}$ is the reward and $\gamma$ is the discount factor.

Here, $\mathcal{N}$ includes $N-1$ *Execution* LLMs and a *Coordinator* LLM to achieve belief coordination. Each Execution LLM maintains a local history $\tau_i^t = \{a_i^1, o_i^1, \ldots, a_i^{t-1}, o_i^{t-1}\}$, composed of its previous actions and observations for belief state updates. Execution LLM receives three inputs: the question, a format and strategy from the coordinator, and the final aggregated output from Coordinator LLM. Our objective is to identify a policy profile $\pi = (\pi_1, \ldots, \pi_N)$ forming a BNE through belief coordination, each agent's policy $\pi_i : \mathcal{H}_i \to \Delta(\mathcal{A}_i)$ maps its history to an action distribution based on its belief about other agents' strategies, such that no agent can improve its response quality unilaterally.

## 2.2. BNE Existence and Convergence

To bridge our DEC-POMDP formulation with BNE analysis, we introduce the concept of agent *types*. In our framework, each agent $i$ is associated with a *type* $\theta_i = \tau_i^t$, which is precisely the agent's local history, capturing its internal beliefs and private observation history. Given the partial observability in our DEC-POMDP, each agent forms *beliefs about other agents' types* based on a common prior and its own observations. These beliefs are operationalized through belief networks in our implementation (detailed in Sec. 3).

Formally, a strategy profile $\{\pi_i^*\}_{i=1}^N$ is a BNE if, for each $i$:

$$\mathbb{E}_{\theta_{-i}}\Big[U_i\big(\pi_i^*(\theta_i), \pi_{-i}^*(\theta_{-i}), \theta_i, \theta_{-i}\big)\Big]$$

$$\geq \mathbb{E}_{\theta_{-i}}\Big[U_i\big(\pi_i'(\theta_i), \pi_{-i}^*(\theta_{-i}), \theta_i, \theta_{-i}\big)\Big], \quad \forall \pi_i',$$

where $\theta_{-i} = (\theta_1, \dots, \theta_{i-1}, \theta_{i+1}, \dots, \theta_N)$ denotes the types of all agents except $i$, utility function is defined as:

$$U_i(\pi_i^*, \pi_{-i}^*, \theta_i, \theta_{-i}) = \mathbb{E}\left[\sum_{t=0}^{\infty} \gamma^t r_i^t \mid \pi_i^*, \pi_{-i}^*, \theta_i, \theta_{-i}\right],$$

where $r_i^t = \mathcal{R}(s^t, a^t)_i$ is agent $i$'s reward from the DEC-POMDP reward function at time $t$. To establish the existence of BNE in our setting, we verify three standard conditions: (1) each agent's mixed strategy space is non-empty, compact, and convex (satisfied by our bounded prompt embedding space); (2) the payoff function $U_i(\theta, a)$ is continuous in types and actions (ensured by continuous reward functions); and (3) each agent's expected payoff is quasi-concave in its own actions for fixed $\theta_i$ (guaranteed by our reward design).

**Theorem 2.1** (Existence of Bayesian Nash Equilibrium)**.** *In this multi-agent LLM framework, if the above conditions are satisfied, there exists a BNE strategy profile $\overline{\pi}^* = (\pi_1^*, \dots, \pi_N^*)$ by Glicksberg's Fixed Point Theorem (Ahmad et al., 2023). A full proof is provided in Appendix B.1.*

We then analyze the convergence properties of our ECON framework via Bayesian regret. Our analysis shows that combining belief networks with coordinated updates leads to an efficient approach for approximating BNE, achieving a sublinear regret bound $O\big(N\sqrt{T}/1-\gamma\big)$, in contrast to the linear regret of existing multi-agent debate methods.

To connect our theoretical analysis with the DEC-POMDP formulation, we define the value function and regret in terms of our system components. For each agent $i$, we measure learning efficiency using Bayesian regret over $T$ steps:

$$R_i(T) = \mathbb{E}_{s_t, \pi_t}\left[\sum_{t=1}^T \big(V_i^*(s_t) - V_i^{\pi_t}(s_t)\big)\right],$$

where $V_i^*(s)$ denotes optimal value function under BNE:

$$V_i^*(s) = \max_{\pi_i} \mathbb{E}_{\pi_{-i}^*}\left[\sum_{t=0}^{\infty} \gamma^t \mathcal{R}(s^t, a^t)_i \mid s^0 = s, \pi_i, \pi_{-i}^*\right],$$

and $V_i^{\pi_t}(s)$ is the value under the current policy profile $\pi_t = (\pi_1^t, \dots, \pi_N^t)$ at time $t$. The expectation accounts for randomness in both state transitions (governed by $\mathcal{P}$) and policy choices. To analyze the total Bayesian regret $R(T) = \sum_{i=1}^N R_i(T)$, we impose standard assumptions (see Appendix B.3) and propose Lemma 2.2, proven in Appendix C.1. Using Lemma 2.2, we bound the Bayesian regret and provide a proof sketch below, with more details and a bound comparison with MAD in Appendices C.2–C.3.

**Lemma 2.2** (Performance Difference)**.** *For joint policies $\pi = (\pi_i, \pi_{-i})$ and $\pi' = (\pi_i', \pi_{-i}')$, the difference in value:*

$$V_i^{\pi'}(s) - V_i^{\pi}(s) = \frac{1}{1-\gamma} \mathbb{E}_{s \sim d_{\pi'}}\Big[\mathbb{E}_{a \sim \pi'} Q_i^{\pi}(s, a)$$
$$- \mathbb{E}_{a \sim \pi} Q_i^{\pi}(s, a)\Big],$$

*where $d_{\pi'}$ is the state distribution under $\pi'$, and $a = (a_i, a_{-i})$ denotes the joint action from $\mathcal{A}$.*

Note that the Q-function $Q_i^{\pi}(s, a)$ appearing in this lemma will be approximated by neural networks in our implementation, as detailed in Sec. 3. Applying this lemma to our regret analysis yields (Jin et al., 2020; Fujimoto et al., 2018):

$$R(T) = \sum_{i=1}^N \frac{1}{1-\gamma} \mathbb{E}_{s_t, \pi_t}\Big[\sum_{t=1}^T \big(\mathbb{E}_{a_t^* \sim \pi^*} Q_i^{\pi_t}(s_t, a_t^*)$$
$$- \mathbb{E}_{a_t \sim \pi_t} Q_i^{\pi_t}(s_t, a_t)\big)\Big],$$

where $\pi^*$ represents the BNE policies. By bounding the estimation error $\epsilon_t$ and suboptimality $\delta_t$, we show

$$\mathbb{E}_{a_t^* \sim \pi^*} Q_i^{\pi_t}(s_t, a_t^*) - \mathbb{E}_{a_t \sim \pi_t} Q_i^{\pi_t}(s_t, a_t) \leq 2\epsilon_t + \delta_t,$$

where $\epsilon_t = O(1/\sqrt{t})$ bounds the Q-function estimation error and $\delta_t = O(1/\sqrt{t})$ measures policy suboptimality. Under standard regularity conditions, these errors can be bounded by constants $C_\epsilon$ and $C_\delta$ respectively, leading to

$$R(T) \leq \sum_{i=1}^N \frac{1}{1-\gamma}\big(2C_\epsilon + C_\delta\big) \sum_{t=1}^T \frac{1}{\sqrt{t}} = O\Big(\frac{N\sqrt{T}}{1-\gamma}\Big).$$

## 3. BNE Implementation with ECON

We present ECON that satisfies the DEC-POMDP architecture in this section. Our framework satisfies the assumptions in Appendix B.3, enabling the application of Lemma 2.2 to bound Bayesian regret. As illustrated in Figure 2, ECON adopts a *Coordinator-Executor* architecture where multiple Execution LLMs operate locally under the guidance of a Coordinator LLM, and comprises two phases: *Inference* and *Optimization*. Below we first introduce the core modules that collectively implement our DEC-POMDP design, then describe the complete training process in Algorithm 1.

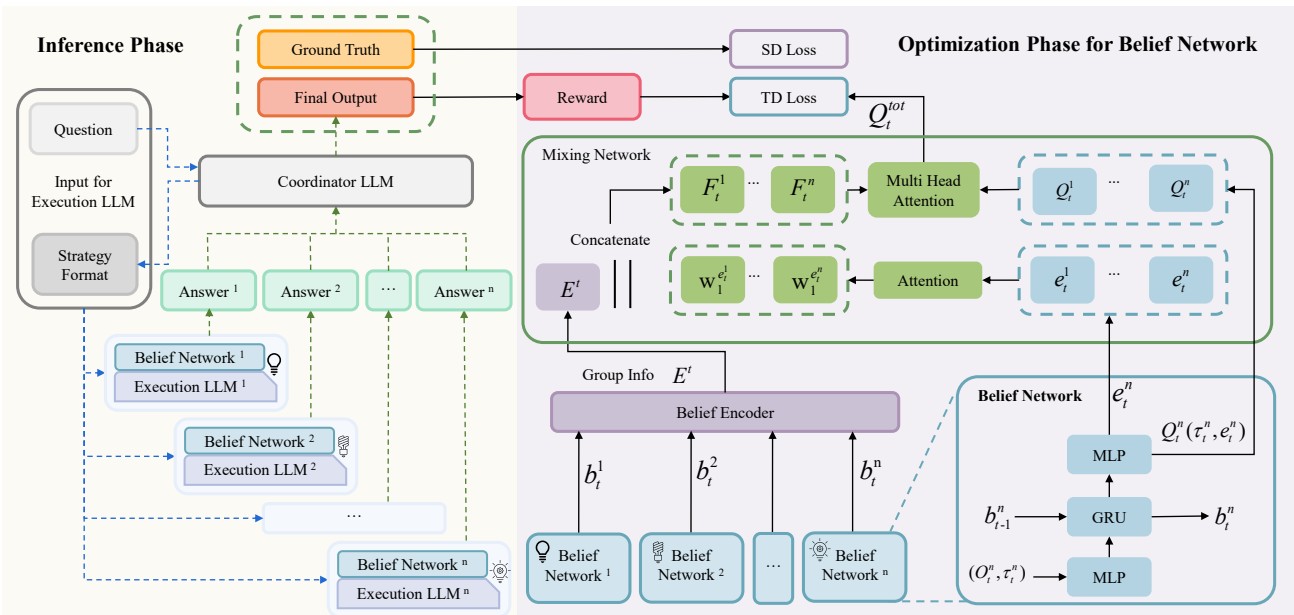

Figure 2: ECON Framework: The *inference* procedure (left) shows how the Coordinator LLM processes and manages Execution LLMs' responses. The *optimization* procedure (right) illustrates the parameter process of the belief network.

## 3.1. Inference Phase

During the inference phase of ECON, a Coordinator LLM generates an informative strategy and a format based on the input question. These are then disseminated to the Execution LLMs, which independently produce their respective answers. Then Coordinator LLM aggregates these answers to form a final output, as illustrated in the left part of Figure 2. A detailed case study demonstrating this inference process is provided in Appendix F. To implement the theoretical policy $\pi_i : \mathcal{H}_i \rightarrow \Delta(\mathcal{A}_i)$ from DEC-POMDP formulation, each Execution LLM maintains a belief network that maps its local history to actions. The following subsections detail how ECON collectively approximates towards BNE.

## 3.2. Individual Belief Network

Each Execution LLM $i$ maintains a belief network $B_i(\tau_i^t, o_i^t; \theta_i^B)$ that implements its policy $\pi_i$ by mapping local trajectory $\tau_i^t$ and current observation $o_i^t \in \mathcal{O}_i$ to a belief state $\mathbf{b}_i \in \mathbb{R}^d$. The belief state captures the agent's understanding of the environment and other agents' behaviors under partial observability, enabling strategic decision-making without direct access to others' outputs. The belief state $\mathbf{b}_i$ is further used to generate the action $a_i^t \in \mathcal{A}_i$, represented as prompt embedding $\mathbf{e}_i = [T_i, p_i]$. While $\mathbf{e}_i$ is a 2-dimensional vector, we term it "embedding" as it embeds the agent's strategic decisions into the prompt control space, influencing the LLM's generation. We define:

$$T_i = T_{\min} + (T_{\max} - T_{\min}) \cdot \sigma(W_T \mathbf{b}_i + b_T),$$
$$p_i = p_{\min} + (p_{\max} - p_{\min}) \cdot \sigma(W_p \mathbf{b}_i + b_p).$$

where $\sigma(\cdot)$ is the sigmoid function. Here, $T_i$ modulates the temperature of token sampling, and $p_i$ sets a penalty threshold for repetition. The belief network $B_i$ has two outputs: (1) the prompt embedding $\mathbf{e}_i$ that serves as the action in our DEC-POMDP framework, and (2) a local Q-value $Q_i^t(\tau_i^t, \mathbf{e}_i^t; \phi_i)$ that estimates the expected return from the current belief state. The belief state $\mathbf{b}_i$ is also passed to the belief encoder for group-level processing.

To optimize the belief network, we apply a TD loss to update the parameters $\theta_i^B$, as illustrated in the right part of Figure 2. The belief network parameters $\theta_i^B = \{\phi_i, W_T, b_T, W_p, b_p\}$ include Q-value function parameters $\phi_i$ and prompt embedding parameters. Thus, we have:

$$\mathcal{L}_{\text{TD}}^i(\theta_i^B) = \mathbb{E}_{\mathcal{D}} \left[ \left( r_i^t + \gamma \max_{\mathbf{e}_i^{t+1}} Q_i^{t+1}(\tau_i^{t+1}, \mathbf{e}_i^{t+1}; \phi_i') \right. \right.$$
$$\left. \left. - Q_i^t(\tau_i^t, \mathbf{e}_i^t; \phi_i) \right)^2 \right],$$

where $r_i^t = \mathcal{R}(s^t, a^t)_i$ is the local reward signal and $\phi_i'$ denotes target network parameters (updated via soft update). By minimizing $\mathcal{L}_{\text{TD}}^i$, Execution LLM $i$ refines its belief state to improve local decision making.

## 3.3. Belief Encoder

A shared *belief encoder*, $f_e(\cdot; \theta_e)$, aggregates the belief states from all agents to produce a group-level representation $\mathbf{E} = f_e(\{\mathbf{b}_i\}_{i=1}^N; \theta_e)$. This encoder enables agents to implicitly coordinate their beliefs, facilitating convergence to BNE. We employ multi-head attention with $H$ heads to

capture inter-agent dependencies in belief states:

$$\text{head}_h = \text{Attention}\big(W_h^Q\,\mathbf{b},\,W_h^K\,\mathbf{b},\,W_h^V\,\mathbf{b}\big),$$

where $\mathbf{b} = [\mathbf{b}_1;\ldots;\mathbf{b}_N] \in \mathbb{R}^{Nd}$ concatenates $\{\mathbf{b}_i\}_{i=1}^N$. The final output is $\mathbf{E} = \text{Concat}(\text{head}_1,\ldots,\text{head}_H)\,W^O$, with $\{W_h^Q, W_h^K, W_h^V\}$ as learnable parameters and $W^O$ as the output projection. The belief encoder can also be regularized via $\mathcal{L}_e(\theta_e) = \mathcal{L}_{\text{TD}}^{\text{tot}}(\phi) + \lambda_e \sum_{i=1}^N \mathcal{L}_{\text{TD}}^i(\theta_i^B)$, where $\mathcal{L}_{\text{TD}}^{\text{tot}}$ is the global TD objective. This encoder captures higher-level dynamics among Execution LLMs for coherence.

### 3.4. Centralized Mixing Network

The *Centralized Mixing Network* coordinates the integrated belief information from all Execution LLMs, facilitating global optimization toward BNE. Specifically, each agent's prompt embedding $\{\mathbf{e}_i^t\}_{i=1}^N$ is processed via self-attention to capture agent-agent dependencies, yielding intermediate embeddings $\{\mathbf{w}_i^t\}_{i=1}^N$. We then combine $\{\mathbf{w}_i^t\}_{i=1}^N$ with the group-level representation $\mathbf{E}^t$ to produce feature transformations $\{F_i^t\}_{i=1}^N$. This combination allows the network to integrate individual strategic decisions (captured in $\mathbf{e}_i^t$) with collective belief dynamics (captured in $\mathbf{E}^t$), enabling coordinated optimization. The local Q-values $\{Q_i^t\}_{i=1}^N$ and the transformed features $\{F_i^t\}_{i=1}^N$ are jointly fed into multi-head attention layers to compute a global Q-value $Q_{\text{tot}}^t$. This global Q-value function accounts for local-global interactions, ensuring that improvements in individual behavior also contribute. To train the mixing network, we minimize:

$$\mathcal{L}_{\text{mix}}(\phi) = \mathcal{L}_{\text{TD}}^{\text{tot}}(\phi) + \mathcal{L}_{\text{SD}} + \lambda_m \sum_{i=1}^N \|Q_i^t - Q_{\text{tot}}^t\|^2,$$

where $\mathcal{L}_{\text{TD}}^{\text{tot}}(\phi)$ aligns $Q_{\text{tot}}^t$ with the global reward $r_{\text{tot}}$:

$$\mathcal{L}_{\text{TD}}^{\text{tot}}(\phi) = \mathbb{E}_{\mathcal{D}}\Big[\big(r_{\text{tot}} + \gamma \max_{\{\mathbf{e}_i^{t+1}\}_{i=1}^N} Q_{\text{tot}}^{t+1}(\tau_{t+1}, \{\mathbf{e}_i^{t+1}\}_{i=1}^N; \phi') \\ - Q_{\text{tot}}^t(\tau_t, \{\mathbf{e}_i^t\}_{i=1}^N; \phi))^2\Big],$$

while a similarity difference loss $\mathcal{L}_{\text{SD}}$ (e.g. $\lambda_b \sum_{i=1}^N (1 - \text{sim}(F_i^t, C))^2$) aligns agent features with the coordinator's final output $C$. $\|Q_i^t - Q_{\text{tot}}^t\|^2$ ensures local Q-values remain consistent with the global estimate. Target parameters $\phi'$ are updated by a soft update rule $\phi' \leftarrow \tau\phi + (1 - \tau)\phi'$. As a result, the mixing network optimizes local policies to improve the global objective, promoting stable convergence (see Appendix B.5 for a monotonicity proof).

### 3.5. Reward Design

The reward function $\mathcal{R}_{\text{design}}$ consists of three components, all bounded by $R_{\max}$ per Assumption B.2. The Action Likelihood Reward $r_i^{\text{AL}} = \min(R_{\max}, \text{sim}(u_i, C))$ measures final output consistency via cosine similarity

$\text{sim}(u_i, C) = \frac{u_i \cdot C}{\|u_i\|\|C\|}$ (Zhu et al., 2023). The Task-Specific Reward $r_i^{\text{TS}} = \min(R_{\max}, \text{eval}(u_i, \text{task}))$ evaluates domain-specific performance through normalized scoring (Hao et al., 2023). The Collaborative Contribution Reward $r_i^{\text{CC}} = \min(R_{\max}, \text{quality}(u_i, \{u_j\}_{j \neq i}))$ assesses each agent's contribution to the collective solution (Xie et al., 2024b). The total reward is computed as $r_i = \alpha_1 r_i^{\text{AL}} + \alpha_2 r_i^{\text{TS}} + \alpha_3 r_i^{\text{CC}}$, where $\alpha_1 + \alpha_2 + \alpha_3 = 1$. These weights are dynamically adjusted via gradient updates $\alpha_k \leftarrow \alpha_k - \eta_\alpha \cdot \partial \mathcal{L}_{\text{dr}}/\partial \alpha_k$, with $\mathcal{L}_{\text{dr}} = \sum_{i=1}^N (r_i^{\text{actual}} - r_i^{\text{expected}})^2$.

### 3.6. Early Stopping

To ensure efficient optimization and convergence to stable solutions, early stopping is implemented based on three key criteria. First, final output stability is achieved when the change in the coordinator's output satisfies $\|\Delta C\| = \|C_{t+1} - C_t\| \leq \epsilon_C$. Second, reward convergence is monitored such that the average reward across agents reaches a predefined threshold, $\frac{1}{N} \sum_{i=1}^N r_i \geq R_{\text{threshold}}$. Lastly, loss convergence is ensured when the total loss stabilizes, satisfying $|L_{\text{tot}}^{t+1} - L_{\text{tot}}^t| \leq \epsilon_L$, where $L_{\text{tot}}$ is the sum of individual agent losses $\sum_{i=1}^N L_i$, execution loss $L_e$, and the mixing loss $L_{\text{mix}}$. These criteria monitor the optimization process, ensuring both strategic alignment and performance while preventing premature termination.

---

**Algorithm 1** Belief Network Training Algorithm

---

**Require:** Question, $\{\text{ExecLLM}_i\}_{i=1}^N$, Coordinator LLM, thresholds $\{\epsilon_C, R_{\text{th}}, \epsilon_L\}$
**Ensure:** Optimized parameters $\{\theta_i^B\}_{i=1}^N, \theta_e, \phi$
1: **Initialize:** $\{\theta_i^B\}_{i=1}^N, \theta_e, \phi$
2: **if not converged then**
3:     **for** each LLM $i$ **do**
4:         $\mathbf{b}_i \leftarrow B_i(\tau_i, o_i; \theta_i^B)$
5:         $\mathbf{e}_i \leftarrow \text{ComputeEmbedding}(\mathbf{b}_i)$
6:         $u_i \leftarrow \text{ExecLLM}_i(\text{query}, \mathbf{e}_i)$
7:         $r_i = \alpha_1 r_i^{\text{AL}} + \alpha_2 r_i^{\text{TS}} + \alpha_3 r_i^{\text{CC}}$
8:     **end for**
9:     $\mathbf{E} \leftarrow f_e(\{\mathbf{b}_i\}_{i=1}^N; \theta_e)$
10:    $C \leftarrow \text{Coordinator}(\{u_i\}_{i=1}^N, \mathbf{E})$
11:    Update $\{\theta_i^B\}_{i=1}^N$ via $\mathcal{L}_{\text{TD}}^i$; $\theta_e$ via $\mathcal{L}_e$; $\phi$ via $\mathcal{L}_{\text{mix}}$
12:    converged $\leftarrow \|C - C_{\text{prev}}\| \leq \epsilon_C$ **and** $R_{\text{avg}} \geq R_{\text{th}}$ **and** $|\Delta L_{\text{tot}}| \leq \epsilon_L$
13: **end if**
14: **return** $\{\theta_i^B\}_{i=1}^N, \theta_e, \phi$

---

## 4. Experiment

In this section, we present the experiment setup in Sec. 4.1, demonstrate the method effectiveness in Sec. 4.2, validate the heterogeneous results in Sec. 4.3, test scale-up capability in Sec. 4.4, and conduct ablation studies in Sec. 4.5.

### 4.1. Setups

#### 4.1.1. MODELS AND DATASETS.

We evaluate 6 released opensourced LLMs: LLaMA3.1 8B (Dubey et al., 2024), LLaMA3.1 70B, Mistral-7B (Jiang et al., 2023), LLaMA3.1 405B, Mixtral-8x22B (Jiang et al., 2024) and Qwen1.5 110B (Yang et al., 2024) across 5 reasoning tasks, including 4 mathematical datasets (GSM8K (Cobbe et al., 2021), GSM-Hard (Gao et al., 2023), MATH (Hendrycks et al., 2021), SVAMP (Patel et al., 2021)) and one commonsense reasoning dataset (StrategyQA (Geva et al., 2021)). Then, we evaluate GPT4 turbo) (Achiam et al., 2023) in a very challenging planning task (Travelplanner (Xie et al., 2024a)) to further validate the performance. The details of benchmarks can be found in Appendix C.5.

#### 4.1.2. COMPARED METHODS AND BENCHMARKS.

We compare ECON against several strong baseline types widely adopted: (i) single-round CoT prompting, including zero-shot and few-shot CoT (Kojima et al., 2022; Wei et al., 2022); (ii) multi-round CoT prompting, Self Consistency (SC) (Wang et al., 2023) method, where we sample answers 64 times and employ majority voting for answer selection; (iii) value-guided search approaches with learned action-value functions, including TS-LLM (Feng et al., 2023) which leverages AlphaZero-style value networks for MCTS, and PPO-MCTS (Liu et al., 2024a) which learns value models to evaluate generation quality in tree search; (iv) multi-round self-improving approaches, using ToT (Yao et al., 2023), RAP (Hao et al., 2023) and React (Yao et al., 2022), with BFS and MCTS for tree search, respectively, following their original answer selection; and (v) multi-LLM reasoning frameworks, including rStar (Qi et al., 2024) and multi-agent debate (Du et al., 2024).

#### 4.1.3. ECON SETUPS.

In this section, the ECON framework includes one Coordinator and three Execution LLMs. The hyperparameters for training can be found in Appendix C.6. To ensure a fair comparison with the baseline, we use four identical models for these LLMs. For heterogeneous results, we also evaluate ECON with different models in Table 2. All evaluations are conducted in a zero-shot setting, with a general prompt provided in Appendix E. Notably, while we set a 50-token constraint for the coordinator's strategy generation, considering that LLMs may not strictly follow length instructions (Yuan et al., 2024b), who showed that $95\%$ of responses stay within $1.4\times$ and $50\%$ within $1.0\times$ of the specified length, we implement a 70-token hard cutoff with regeneration mechanism to controls the token usage.

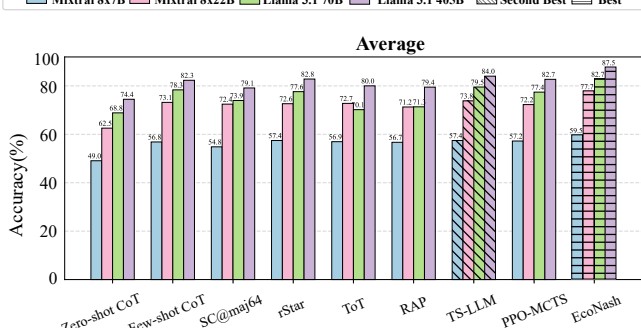

Figure 3: Average results of five reasoning datasets: GSM8K, GSM-Hard, SVAMP, Strategy QA and MATH.

### 4.2. Main Result

Figure 3 presents the average accuracy comparison of each method across four mathematical reasoning datasets and one commonsense reasoning dataset. Detailed accuracy comparisons for each dataset can be found in Appendix E. The empirical results demonstrate that ECON outperforms most baselines on all complex reasoning benchmarks. On average, ECON outperforms the single-round method Zero-shot CoT by $25.6\%$, Few-shot CoT by $6.3\%$, multi-round CoT prompting SC by $10.9\%$, multi-round self-improving approaches ToT by $11.2\%$, and multi-LLM reasoning frameworks rStar by $6.4\%$.

Furthermore, when evaluated on the very challenging Travelplanner benchmark using GPT-4-Turbo in Table 1, ECON enhanced the final pass rates to $7.2\%$ on the validation set and $9.3\%$ on the test set, while compared to $2.3\%$ and $3.7\%$ achieved by a three-round multi-agent debate.

These results demonstrate that ECON effectively leverages the capabilities of more powerful models and outperforms alternative reasoning optimization methods in complex tasks. Additionally, we provide a corresponding example for MATH, which are available in Appendix F. Note that ECON uses fewer tokens compared to multi-round CoT prompting SC, multi-round self-improving approaches ToT, and MAD, meanwhile achieved performance improvements.

### 4.3. Model Configuration and Cost Efficiency Analysis

To evaluate the impact of both the Coordinator LLM and Execution LLM performance on the ECON framework and find whether heterogeneous Execution LLMs can also achieve a BNE, we conducted two types of experiments: one pairing a strong Coordinator LLM with weaker Execution LLMs, and another pairing a weak Coordinator LLM with stronger Execution LLMs. These experiments were further divided into homogeneous and heterogeneous execution groups for detailed analysis. To ensure a fair comparison, the Coordinator LLM was consistently set to LLaMA3.1 70b across all experiments. For the heterogeneous execution group, we

Table 1: Empirical results on the TravelPlanner dataset, along with some leaderboard rankings, are presented.

| | Validation (#180) | | | | | | Test (#1,000) | | | | | |
|---|---|---|---|---|---|---|---|---|---|---|---|---|
| | Delivery Rate | Commonsense Pass Rate | | Hard Constraint Pass Rate | | Final Pass Rate | Delivery Rate | Commonsense Pass Rate | | Hard Constraint Pass Rate | | Final Pass Rate |
| | | Micro | Macro | Micro | Macro | | | Micro | Macro | Micro | Macro | |
| Greedy Search | 100 | 74.4 | 0 | 60.8 | 37.8 | 0 | 100 | 72.0 | 0 | 52.4 | 31.8 | 0 |
| | | | | | Two-stage | | | | | | | |
| Mixtral-8x7B-MoE | 49.4 | 30.0 | 0 | 1.2 | 0.6 | 0 | 51.2 | 32.2 | 0.2 | 0.7 | 0.4 | 0 |
| Gemini Pro | 28.9 | 18.9 | 0 | 0.5 | 0.6 | 0 | 39.1 | 24.9 | 0 | 0.6 | 0.1 | 0 |
| GPT-3.5-Turbo | 86.7 | 54.0 | 0 | 0 | 0 | 0 | 91.8 | 57.9 | 0 | 0.5 | 0.6 | 0 |
| GPT-4-Turbo | 89.4 | 61.1 | 2.8 | 15.2 | 10.6 | 0.6 | 93.1 | 63.3 | 2.0 | 10.5 | 5.5 | 0.6 |
| Debate (GPT-4) @3round | 95.2 | 67.3 | 6.7 | 22.7 | 13.1 | 2.3 | 97.8 | 72.4 | 11.3 | 17.4 | 12.1 | 3.7 |
| **ECON (GPT-4)** | **100** | **71.4** | **15.6** | **32.1** | **25.7** | **7.2** | **100** | **82.1** | **26.6** | **32.4** | **17.6** | **9.3** |
| | | | | | Sole-planning | | | | | | | |
| Direct$_{GPT-3.5-Turbo}$ | 100 | 60.2 | 4.4 | 11.0 | 2.8 | 0 | 100 | 59.5 | 2.7 | 9.5 | 4.4 | 0.6 |
| CoT$_{GPT-3.5-Turbo}$ | 100 | 66.3 | 3.3 | 11.9 | 5.0 | 0 | 100 | 64.4 | 2.3 | 9.8 | 3.8 | 0.4 |
| ReAct$_{GPT-3.5-Turbo}$ | 82.2 | 47.6 | 3.9 | 11.4 | 6.7 | 0.6 | 81.6 | 45.9 | 2.5 | 10.7 | 3.1 | 0.7 |
| Reflexion$_{GPT-3.5-Turbo}$ | 93.9 | 53.8 | 2.8 | 11.0 | 2.8 | 0 | 92.1 | 52.1 | 2.2 | 9.9 | 3.8 | 0.6 |
| Direct$_{Mixtral-8x7B-MoE}$ | 100 | 68.1 | 5.0 | 3.3 | 1.1 | 0 | 99.3 | 67.0 | 3.7 | 3.9 | 1.6 | 0.7 |
| Direct$_{Gemini Pro}$ | 93.9 | 65.0 | 8.3 | 9.3 | 4.4 | 0.6 | 93.7 | 64.7 | 7.9 | 10.6 | 4.7 | 2.1 |
| Direct$_{GPT-4-Turbo}$ | **100** | 80.4 | 17.2 | 47.1 | 22.2 | 4.4 | **100** | 80.6 | 15.2 | 44.3 | 23.1 | 4.4 |
| Debate (GPT-4) | 97.7 | 78.9 | 15.6 | 43.3 | 20.6 | 6.7 | 98.2 | 79.5 | 18.8 | 41.7 | 22.9 | 7.1 |
| **ECON (GPT-4)** | **100** | **83.3** | **22.2** | **51.7** | **27.8** | **12.9** | **100** | **84.2** | **23.5** | **49.8** | **28.7** | **15.2** |

Table 2: Performance of different configurations.

| Method | GSM-Hard | MATH |
|---|---|---|
| **Baselines** | | |
| ECON | 51.43 | 81.47 |
| LLaMA 3.1 7B (Few-shot CoT) | 42.23 | 62.71 |
| **ECON Configurations** | | |
| Homo. (3× LLaMA3.1 8B) | 48.71 | 67.70 |
| Homo. (3× LLaMA3.1 405B) | 61.29 | 89.24 |
| Hetero. (LLaMA3.1 8B, LLaMA3 8B, Mixtral 7B) | 45.24 | 74.24 |
| Hetero. (Mixtral 8×22B, Qwen1.5 110B, LLaMA3.1 405B) | 55.73 | 85.46 |

used the following configurations: LLaMA 3.1 8b, LLaMA 3 8b, and Mixtral 7b, as well as another configuration consisting of Mixtral 8×22b, Qwen1.5 110b, and LLaMA3.1 405b. For the homogeneous execution group, two configurations were tested: one with three weak models (LLaMA 3.1 8b), and another with three strong models LLaMA 3.1 405b. Experimental results indicate that stronger Execution LLMs improve performance by providing higher-quality answers and achieving BNE more efficiently. Additionally, heterogeneous model perform worse than homogeneous models due to increased challenges in reaching BNE, but still outperform baseline Few-shot CoT .

To assess the cost efficiency of the ECON framework, Table 3 presents the average token usage of ECON, Multi-Agent Debate, RAP, and SC across the MATH, GSM8K, and GSM-Hard datasets for three models: LLaMA 3.1 70B, Mixtral 8x7B, and Mixtral 8x22B. The results demonstrate that ECON reduces token consumption by an average of 21.4% compared to Multi-Agent Debate (3 rounds). Notably, when the Coordinator LLM provides detailed strategies with answer(as shown in the token consumption data in Table 3), token usage increases an average of 112% higher as the full strategy processing.

We would like to point out that ECON does not eliminate inter-agent communication entirely, but rather adopts an incomplete-information perspective that minimizes communication. ECON's optimization objective focuses on achieving consensus among Execution LLMs, consensus achieved as the result of implicit communication. We make an additional experiment to demonstrate incorporating direct complete interaction into ECON (which makes it become a complete information formation). The performance improve 1.1% while the token consumption increase 42.4% .

### 4.4. Scale Up Result

We analyzed the impact of varying the number of agents further to validate ECON across a broader range of LLMs. We conducted three sets of experiments on the MATH, GSM-Hard, SVAMP, and StrategyQA datasets, aiming to address three key questions: (1) To what extent can weaker LLMs be enhanced? (examined on LLaMA 3.1 8B), (2) Can stronger LLMs be further improved? (using LLaMA 3.1 70B), and (3) Should the number of Coordinator LLMs be increased along with the number of Execution LLMs? Starting from three Execution LLMs (as in the main results), we gradually increased the number of agents to nine. We used the few-shot CoT as the baseline (in grey line) as Figure 4. The results suggest that beyond four Execution LLMs, performance improvements were minimal, and in some cases, performance even declined. We attribute this to the challenge faced by the Coordinator LLM in managing an excessive number of Execution LLMs, making it difficult to achieve coordination by information from additional agents.

Instead of simply increasing the number of Execution LLMs, we enhance scalability by forming a global Nash Equilib-

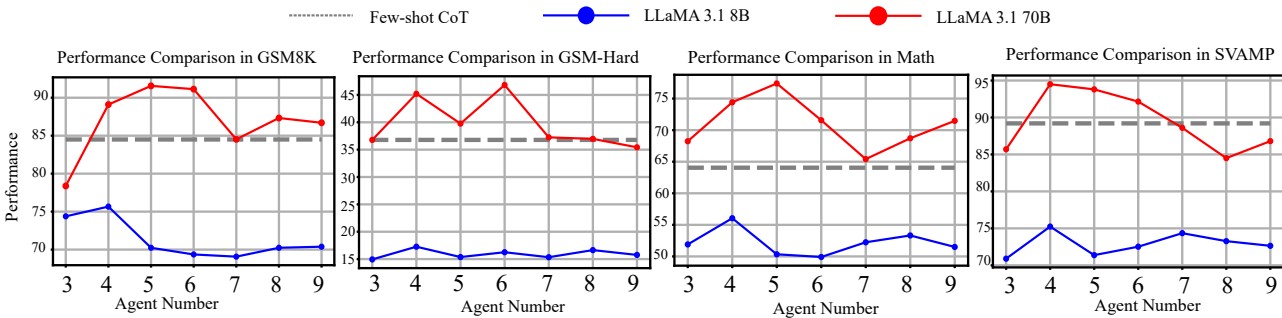

Figure 4: Scaling up our framework with a single coordinator while increasing the number of Execution LLMs in 5 datasets.

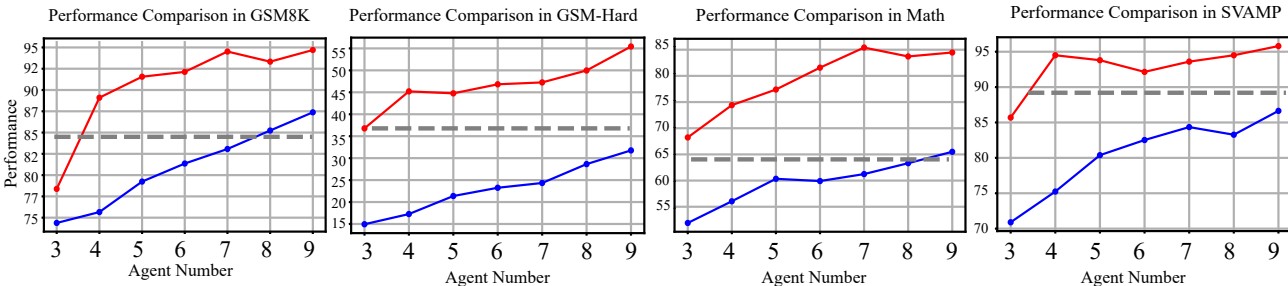

Figure 5: Scaling up by increasing the number of coordinators in proportion to the number of Execution LLMs in 5 datasets.

rium through local Nash equilibria by introducing additional coordinators. This setup ensures that each Coordinator handles a reasonable amount of data. Specifically, each Coordinator manages up to 4 Execution LLMs, forming outputs and guiding them toward local Nash equilibria. Furthermore, a central LLM was introduced to coordinate the multiple coordinators, facilitating the transition from local Nash equilibria to a global Nash Equilibrium (details in Appendix 2). We observed significant improvements across all benchmarks, both for weaker models (LLaMA 3.1 8B) and stronger models (LLaMA 3.1 70B). Compared to a system with 3 Execution LLMs and one coordinator, the scaled-up system with 9 Execution LLMs, 3 coordinators, and a central LLM achieved 18.1% improvement in Figure 5, which has potential to further scale up.

### 4.5. Ablation Study

In the additional experiments, heterogeneous Execution LLMs experienced a slight performance decline. An intuitive explanation for this observation is that implementing BNE is more challenging for heterogeneous Execution LLMs. To verify the actual performance differences of the ECON framework before and after achieving BNE, we conducted experiments on three math reasoning benchmarks: GSM8K, GSM-Hard, and MATH. Results in Table 5 demonstrate that our framework achieved an average performance improvement of 14% after implementing BNE.

Additionally, we performed ablation studies on various submodules, including the reward design and the setting where

the Coordinator LLM provides a strategy without giving a direct answer, to ensure the validity of our architecture. All experiments were conducted with LLaMA 3.1 70B, tested on the MATH benchmark. Specifically, $R_1$ refers to the action likelihood reward, $R_2$ to the task-specific reward, and $R_3$ to the self-evaluation reward. $S_1$ represents the setting where the coordinator does not provide any strategy, while $S_2$ represents the setting where the coordinator provides detailed strategy, $S_3$ represents ECON as our baseline.

Theoretically, the design of our mixing network optimizes local policies to improve the global objective, Monotonicity proofs are demonstrated in Appendix A.5. We make additional ablation experiments to validate this by removing the concatenation and removing the belief encoder in Table 6.

## 5. Related Work

**Prompting Large Language Models to Reason.** Large language models are significantly more capable of complex reasoning with the advancement of prompt techniques (Wei et al., 2022; Kojima et al., 2022; Zhou et al., 2022; Fu et al., 2022; Zhang et al., 2023c; Wang et al., 2023; Li et al., 2023; Chia et al., 2023; Cao et al., 2024; Zhou et al., 2024a; Wang et al., 2025; Zhou et al., 2025). Wei et al. (2022) introduced Chain-of-Thought (CoT) prompting, which presents step-by-step reasoning examples within the prompt. This enables the model to engage in explicit reasoning, enhancing its ability to follow the logical progression that leads to the correct answer. Various extensions of CoT have been developed to improve reasoning performance further. Zero-shot

Table 3: Average token usage and performance comparison in the MATH, GSM8K, and GSM-Hard.

| Dataset | Inference Strategy | LLaMA3.1 70B | | Mixtral 8x7b | | Mixtral 8x22b | |
|---------|-------------------|--------------|-------------|--------------|-------------|---------------|-------------|
| | | Token Usage | Performance | Token Usage | Performance | Token Usage | Performance |
| MATH | Multi-Agent Debate (3 rounds) | 2154.87 | 71.58 | 1462.12 | 31.28 | 5345.56 | 67.41 |
| | RAP | 2653.27 | 68.71 | 1737.73 | 33.99 | 6668.55 | 62.53 |
| | ECON (with detailed strategy) | 3270.06 | 72.38 | 2150.23 | 26.18 | 8054.03 | 68.23 |
| | Self Consistency (64 rounds) | 11917.00 | 67.39 | 8066.21 | 31.58 | 29616.13 | 62.21 |
| | ECON | 1629.79 | 81.47 | 1128.23 | 35.02 | 4270.86 | 72.29 |
| GSM8K | Multi-Agent Debate (3 rounds) | 1391.57 | 86.32 | 1463.40 | 70.19 | 5714.05 | 81.95 |
| | RAP | 1907.86 | 81.33 | 1248.66 | 72.03 | 6517.77 | 76.97 |
| | ECON (with detailed strategy) | 2772.24 | 85.17 | 1188.13 | 65.37 | 9341.60 | 81.46 |
| | Self Consistency (64 rounds) | 9574.25 | 89.56 | 6601.34 | 71.08 | 24671.91 | 86.24 |
| | ECON | 1131.65 | 92.70 | 1284.98 | 76.97 | 4715.31 | 88.20 |
| GSM-Hard | Multi-Agent Debate (3 rounds) | 3030.73 | 41.98 | 1478.14 | 20.04 | 9250.78 | 45.21 |
| | RAP | 1768.72 | 38.97 | 1036.11 | 22.47 | 6464.52 | 42.79 |
| | ECON (with detailed strategy) | 3662.64 | 44.12 | 2239.07 | 18.52 | 11464.98 | 41.04 |
| | Self Consistency (64 rounds) | 16724.69 | 39.76 | 11668.19 | 22.47 | 74544.25 | 44.19 |
| | ECON | 1518.76 | 51.43 | 1271.53 | 25.76 | 7101.62 | 47.58 |

Table 4: Comparison between Token Consumption and Efficiency of ECON w/o complete information in LLaMA 3.1.

| Dataset & Model | Complete Info | Consumption |
|-----------------|---------------|-------------|
| GSM8K - LLaMA 3.1 8B | 81.4 | 80.3 (+35.6%) |
| GSM8K - LLaMA 3.1 70B | 96.1 | 96.7 (+42.7%) |
| GSM-Hard - LLaMA 3.1 8B | 30.2 | 29.9 (+62.3%) |
| GSM-Hard - LLaMA 3.1 70B | 53.6 | 51.4 (+40.9%) |
| MATH - LLaMA 3.1 8B | 59.6 | 60.4 (+33.8%) |
| MATH - LLaMA 3.1 70B | 83.1 | 81.5 (+39.4%) |

Table 5: Ablation on different reward and strategy settings.

| Reward | | | | Strategy | | | |
|--------|--------|--------|-------|----------|--------|--------|-------|
| $R_1$ | $R_2$ | $R_3$ | ECON | $S_1$ | $S_2$ | $S_3$ | ECON |
| ✓ | ✗ | ✓ | 77.55 | ✓ | ✗ | ✗ | 71.35 |
| ✓ | ✗ | ✗ | 74.32 | ✗ | ✓ | ✗ | 72.31 |
| ✓ | ✓ | ✗ | 76.21 | ✗ | ✗ | ✓ | 81.47 |
| | Random | | 62.71 | | | | |

Table 6: Ablation Study on Concatenate and Belief Encoder.

| Dataset & Model | No Concatenate | No belief encoder |
|-----------------|----------------|-------------------|
| GSM8K - LLaMA 3.1 8B | 85.9 (-1.8) | 84.1 (-3.4) |
| GSM8K - LLaMA 3.1 70B | 93.6 (-3.1) | 91.1 (-5.6) |
| GSM-Hard - LLaMA 3.1 8B | 24.9 (-5.0) | 21.7 (-8.2) |
| GSM-Hard - LLaMA 3.1 70B | 47.2 (-4.2) | 42.6 (-8.8) |
| MATH - LLaMA 3.1 8B | 55.6 (-4.8) | 52.3 (-7.1) |
| MATH - LLaMA 3.1 70B | 77.0 (-4.4) | 75.3 (-6.2) |

CoT (Kojima et al., 2022) eliminates the need for manually constructing exemplars, prompting models with phrases like "Let's think step by step" to encourage reasoning. Wang et al. (2023) proposed self-consistency (SC) sampling, where multiple reasoning paths are sampled, and the final answer is determined by majority voting. To enable LLMs to engage in deliberate decision-making, Tree of Thoughts (ToT) (Yao et al., 2023) generates multiple potential answers at each reasoning step, building a tree of possible solutions.

**Multi-agent Debate for Large Language Models Reasoning.** Various multi-agent debate strategies (Du et al., 2024;

Chan et al., 2024; Liang et al., 2023; Chen et al., 2023; Smit et al., 2024; Zhang et al., 2023a; Pham et al., 2023; Li et al., 2023) have been developed to strengthen the reasoning ability of LLMs. Du et al. (2024) introduced an approach where multiple instances of LLMs propose their individual reasoning processes, engaging in multiple rounds of debate to reach a consensus on the final answer. This method not only significantly enhances reasoning performance across a variety of tasks but also reduces the occurrence of hallucinations. Some studies (Chan et al., 2024; Liang et al., 2023) incorporate role-playing into multi-agent debate strategies using role-specific prompts, foster divergent thinking and enhance the reasoning capabilities of LLMs. However, current multi-agent debate strategies face high computational costs and lack theoretical guarantees for convergence. In this work, we introduce an incomplete information perspective to enhance the scalability of multiple LLMs to ensure independent reasoning by each Execution LLM, while ensure convergence through rigorous theoretical analysis.

## 6. Conclusion

In this work, we introduce ECON, a novel collaborative reasoning framework in multi-LLM systems. ECON constructs a hierarchical coordination mechanism, enabling multiple Execution LLMs to engage in distributed reasoning guided by a Coordinator LLM. The hierarchical coordination mechanism allows each Execution LLM to operate independently with its own belief network, receiving only the question and strategy from the Coordinator LLM. This enables multiple Execution LLMs to engage in distributed reasoning to achieve BNE. Experimental results across six benchmarks demonstrate ECON outperforms single-agent approaches by 10.9% in average, confirming the robustness and efficiency of our framework. Moreover, ECON demonstrate great potential of scalability while maintain reasonable cost.

## Acknowledgements

ZKZ, CTC, and BH were supported by RGC Young Collaborative Research Grant No. C2005-24Y, NSFC General Program No. 62376235, Guangdong Basic and Applied Basic Research Foundation Nos. 2022A1515011652 and 2024A1515012399, HKBU Faculty Niche Research Areas No. RC-FNRA-IG/22-23/SCI/04, and HKBU CSD Departmental Incentive Scheme. TLL is partially supported by the following Australian Research Council projects: FT220100318, DP220102121, LP220100527, LP220200949, and IC190100031.

## Impact Statement

This paper presents work whose goal is to advance the field of machine learning. There are many potential societal consequences of our work, none of which we feel must be specifically highlighted here.

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

# Appendix

# A. Notations

This section summarizes the key notations used throughout the appendix and the main paper where applicable.

| Notation | Description |
|---|---|
| **General Game Theory and Reinforcement Learning** | |
| $N$ | Number of agents. |
| $i$ | Index for an agent, $i \in \{1, \dots, N\}$. |
| $\pi_i$ | Strategy or policy for agent $i$. |
| $\pi_{-i}$ | Strategies of all agents other than $i$. |
| $\overline{\pi}^*$ | Bayesian Nash Equilibrium (BNE) strategy profile. |
| $\Pi_i$ | Set of all admissible strategies for agent $i$. |
| $a_i$ | Action taken by agent $i$. |
| $a_{-i}$ | Actions taken by all agents other than $i$. |
| $a = (a_i, a_{-i})$ | Joint action profile. |
| $\mathcal{A}_i$ | Action space for agent $i$. |
| $\mathbf{s}, s$ | State of the environment. |
| $U_i(\cdot)$ | Payoff or utility function for agent $i$. |
| $BR_i(\pi_{-i})$ | Best response correspondence for agent $i$ to $\pi_{-i}$. |
| $r_i$ | Reward received by agent $i$. |
| $R_{\max}$ | Uniform upper bound for rewards. |
| $\gamma$ | Discount factor for future rewards, $0 \le \gamma < 1$. |
| $L_i(\theta_i)$ | Loss function for agent $i$ (e.g., TD loss), parameterized by $\theta_i$. |
| $Q_i(\mathbf{s}, a_i; \theta_i)$ | Q-value function for agent $i$ for state $\mathbf{s}$ and action $a_i$, parameterized by $\theta_i$. |
| $Q_i^*(s, a_i)$ | Optimal Q-value function. |
| $\theta_i$ | Parameters of agent $i$'s network (e.g., Q-network, belief network). Also used to denote agent type in BNE proof. |
| $\theta_i^B$ | Parameters of agent $i$'s belief network. |
| $\theta_e$ | Parameters of the coordinator's belief aggregation function. |
| $\theta_i^-$ | Parameters of the target Q-network for agent $i$. |
| $\eta_t, \eta_0, \eta, \eta', \eta_{\text{global}}, \eta_\alpha$ | Learning rates. |
| $\mathcal{D}_i$ | Replay buffer or data distribution for agent $i$. |
| $D_t$ | Historical data up to time $t$. |
| $V_i^\pi(s)$ or $V_t(O_i)$ | Value function for agent $i$ under policy $\pi$ (or at time $t$ for observation $O_i$). |
| $V_i^*(s)$ | Optimal value function for agent $i$. |
| $\mathcal{B}_t$ or $B_t$ | Bellman operator at time $t$. |
| $\mathcal{B}^*$ | True Bellman optimality operator. |
| $R_i(T)$ | Regret for agent $i$ over $T$ time steps. |
| $R(T)$ | Total regret for all agents over $T$ time steps. |
| $T$ | Total number of time steps or episodes. |
| $\epsilon$ | Small positive constant (e.g., exploration rate, error margin, convergence threshold). |
| $D_{\text{KL}}(\cdot \| \cdot)$ | Kullback-Leibler divergence. |
| $\mathbb{E}[\cdot]$ | Expectation operator. |
| $\nabla$ | Gradient operator. |
| **MA-LLM Framework Specifics (ECON)** | |
| $\Theta_i$ | Type space for agent $i$ (in BNE proof). |
| $O_i$ | Observation for agent $i$, typically $O_i = [e_t, e_s, \mathbf{b}_i]^\top$. |
| $e_t$ | Task encoding / global observation. |
| $e_s$ | Coordinator's strategy (embedding). |
| $\mathbf{b}_i$ | Belief state (embedding) for agent $i$. |
| $\mathbf{e}_i$ | Prompt embedding generated by agent $i$ (considered as its action in some contexts). |

| Notation | **Table 7 – continued from previous page** 
 Description |
|---|---|
| $B_i(\tau_i, O_i; \theta_i^B)$ | Belief network function for agent $i$. |
| $\tau_i$ | Historical trajectory or local information for agent $i$. |
| $f_e(\{\mathbf{b}_i\}_{i=1}^N; \theta_e)$ | Coordinator's belief aggregation function. |
| $\mathbf{E}$ | Aggregated group information/embedding from coordinator. |
| $C, C^k$ | Coordinator's final output (embedding). |
| $P_{\text{post}}(\mathbf{E} \mid \cdot)$ | Posterior distribution over group information determined by the coordinator. |
| $P_{\text{LLM}}(\mathbf{E} \mid \cdot)$ | Belief distribution over group information maintained by an Execution LLM. |
| $H_t$ | Belief entropy at time $t$. |
| $I(X; Y \mid Z)$ | Conditional mutual information between random variables $X$ and $Y$ given $Z$. |
| $\xi(\mathbf{e}_i, \mathbf{E})$ | A function related to coordination, depending on $\mathbf{e}_i$ and $\mathbf{E}$. |
| $\eta$ | Game regularity constant (distinct from learning rate). |
| $\kappa$ | Concentrability constant. |
| $u_i$ | Output (e.g., text, solution) generated by Execution LLM $i$. |
| $K$ | Number of clusters in the hierarchical scaling-up framework. |
| $\text{Coord}_{\text{global}}$ | Global Coordinator LLM. |
| $\text{Coord}_k$ | Local Coordinator LLM for cluster $k$. |
| $\mathbf{S}$ | Global strategy (embedding) from $\text{Coord}_{\text{global}}$. |
| $\mathbf{s}_k$ | Local strategy (embedding) for cluster $k$ from $\text{Coord}_k$. |
| $C_k$ | Cluster of Execution LLMs (also used for local output from $\text{Coord}_k$, context dependent). |
| $\epsilon_C, \epsilon_L$ | Convergence thresholds for final output and losses respectively. |
| $R_{\text{threshold}}$ | Reward threshold for early stopping in scaling-up framework. |
| $r_i^{\text{AL}}$ | Action Likelihood Reward. |
| $r_i^{\text{TS}}$ | Task-Specific Reward. |
| $r_i^{\text{CC}}$ | Collaborative Contribution Reward. |
| $\alpha_1, \alpha_2, \alpha_3$ | Weights for combining reward components. |
| $Q_{\text{tot}}$ | Total Q-value from the mixing network. |
| $h^l$ | Hidden state of layer $l$ in the mixing network. |
| $W^l, b^l$ | Weights and biases for layer $l$ of the mixing network. |
| $\phi^l(\cdot)$ | Non-decreasing activation function for layer $l$ of the mixing network. |
| $d_\pi^k$ | State distribution at step $k$ induced by policy $\pi$. |
| $T_i, p_i$ | Temperature and nucleus sampling (top-p) parameters for LLM $i$. |
| $\mathcal{L}_{\text{dr}}$ | Loss for dynamic reward weight adjustment. |

**Proof Specifics (Regret Bounds, Convergence)**

| | |
|---|---|
| $E_1(t), E_2(t)$ | Error terms in Q-value decomposition (related to Q-function estimation error). |
| $\Delta(t)$ | Policy suboptimality term in Q-value decomposition. |
| $\epsilon_t(s, a)$ | Q-function estimation error $|Q_t(s, a) - Q^*(s, a)|$ at time $t$. |
| $C_1, C_2, C$ | Generic constants appearing in convergence rate and regret bounds. |
| $\zeta_t$ or $\xi_t$ | Martingale difference noise term in stochastic approximation proofs. |
| $\sigma^2$ | Bound on the variance of the noise term $\zeta_t$. |
| $\mathcal{P}$ | Policy space (assumed convex and compact in some proofs). |
| $J(\pi)$ | Policy objective function (e.g., expected discounted return of policy $\pi$). |
| $L$ | Lipschitz constant (e.g., for policy gradient). |
| $D(\mathcal{P})$ | Diameter of the policy space $\mathcal{P}$. |
| $\delta_{\min}$ | Lower bound on persistent policy suboptimality in competitive (debate) settings. |
| $v_i$ | Value of the game for agent $i$ (in zero-sum games). |
| $H(\pi_i)$ | Entropy of policy $\pi_i$. |
| $h_{\min}$ | Minimum entropy required for policies in competitive settings to avoid exploitation. |

# B. Theoretical Proof

## B.1. Proof of Theorem 2.1

*Proof.* We aim to prove the existence of a Bayesian Nash Equilibrium (BNE) in our multi-agent LLM framework under the specified conditions. The proof proceeds by verifying the conditions of Glicksberg's Fixed Point Theorem, which guarantees the existence of a fixed point in continuous games with infinite-dimensional strategy spaces.

**Step 1: Define the Best Response Correspondence**

For each agent $i$, define the best response correspondence $BR_i$ as:

$$BR_i(\pi_{-i}) = \{\pi_i \in \Pi_i \mid \pi_i \text{ maximizes } U_i(\theta_i, \pi_i, \pi_{-i})\},$$

where $\Pi_i$ is the set of all admissible strategies for agent $i$, and $\pi_{-i}$ denotes the strategies of all other agents.

**Step 2: Verify the Conditions of Glicksberg's Fixed Point Theorem**

To apply Glicksberg's Fixed Point Theorem, we need to verify the following conditions for each agent $i$:

1. *Strategy Space Compactness and Convexity*:
   - The strategy space $\Pi_i$ is non-empty, convex, and compact in the topology of pointwise convergence.

2. *Continuity of Payoff Functions*:
   - The payoff function $U_i(\theta_i, \pi_i, \pi_{-i})$ is continuous in $(\pi_i, \pi_{-i})$ for each fixed $\theta_i$.

3. *Quasi-Concavity of Payoff Functions*:
   - The payoff function $U_i(\theta_i, \pi_i, \pi_{-i})$ is quasi-concave in $\pi_i$ for each fixed $\theta_i$ and $\pi_{-i}$.

*Verification*:

1. **Strategy Space Compactness and Convexity**:

   The strategy space $\Pi_i$ consists of all measurable functions mapping types $\theta_i$ to actions $a_i$ in $\mathcal{A}_i$. Since $\Theta_i$ and $\mathcal{A}_i$ are compact metric spaces, and strategies are measurable functions from one compact space to another, the space of such functions $\Pi_i$ can be endowed with the topology of pointwise convergence, making it compact by Tychonoff's Theorem. Convexity follows because the set of mixed (probabilistic) strategies is convex, and any convex combination of measurable functions is measurable.

2. **Continuity of Payoff Functions**:

   For fixed $\theta_i$, the payoff function $U_i(\theta_i, \pi_i, \pi_{-i})$ depends continuously on $\pi_i$ and $\pi_{-i}$ due to the continuity of $U_i$ in actions and types. Specifically, since $U_i$ is continuous in $a = (a_i, a_{-i})$ and the strategies $\pi_i, \pi_{-i}$ map continuously from types to actions, the composition $U_i(\theta_i, \pi_i(\theta_i), \pi_{-i}(\theta_{-i}))$ is continuous in $(\pi_i, \pi_{-i})$.

3. **Quasi-Concavity of Payoff Functions**:

   For each $\theta_i$ and $\pi_{-i}$, the function $\pi_i \mapsto U_i(\theta_i, \pi_i, \pi_{-i})$ is quasi-concave because $U_i$ is quasi-concave in $a_i$ and the strategies are linear in the space of mixed strategies. Therefore, any convex combination of strategies does not decrease the utility, satisfying quasi-concavity.

**Step 3: Establish Upper Hemicontinuity and Non-Empty, Convex-Valuedness of Best Response Correspondences**

We need to show that $BR_i(\pi_{-i})$ is upper hemicontinuous with non-empty, convex values.

1. **Non-Empty, Convex Values**:

   For each $\pi_{-i}$, since $\Pi_i$ is compact and convex, and $U_i$ is continuous and quasi-concave in $\pi_i$, the Weierstrass Theorem ensures that the maximum exists; hence, $BR_i(\pi_{-i})$ is non-empty. Convexity follows from the quasi-concavity of $U_i$ in $\pi_i$, implying that any convex combination of best responses is also a best response.

2. **Upper Hemicontinuity**:

Upper hemicontinuity of $BR_i$ means that for any net $\pi^\alpha_{-i} \to \pi_{-i}$, and any $\pi_i \in BR_i(\pi_{-i})$, there exists a net $\pi^\alpha_i \in BR_i(\pi^\alpha_{-i})$ such that $\pi^\alpha_i \to \pi_i$. This property holds because the payoff function $U_i$ is continuous in $(\pi_i, \pi_{-i})$, and the strategy spaces are compact.

### Step 4: Application of Glicksberg's Fixed Point Theorem

Having verified all the conditions, we can apply Glicksberg's Fixed Point Theorem, which states that if each player's strategy set is compact and convex, and their payoff functions are continuous and quasi-concave in their own strategies, then the game has at least one Nash Equilibrium in mixed strategies.

### Step 5: Conclusion

Therefore, there exists a strategy profile $\overline{\pi}^* = (\pi^*_1, \pi^*_2, \dots, \pi^*_N)$ such that for each agent $i$,

$$\pi^*_i \in BR_i(\pi^*_{-i}),$$

meaning that no agent can unilaterally deviate to improve their expected payoff, given their beliefs about other agents' types and strategies. This strategy profile constitutes a Bayesian Nash Equilibrium in our multi-agent LLM framework.

$\square$

### B.2. Proof of Convergence to BNE

*Proof.* We aim to show that, by minimizing the TD loss for each agent's Q-network, the agents' strategies converge to a Bayesian Nash Equilibrium (BNE).

**Assumptions:**

1. The Q-networks $Q_i(\mathbf{s}, a_i; \theta_i)$ are parameterized by prompt embeddings $\theta_i$, and the mapping from $\theta_i$ to $Q_i$ is continuously differentiable.

2. The exploration strategy ensures sufficient coverage of the state-action space (e.g., $\epsilon$-greedy with decaying $\epsilon$).

3. The loss function $L_i(\theta_i)$ is convex or has Lipschitz continuous gradients with respect to $\theta_i$.

4. The gradient $\nabla_{\theta_i} L_i(\theta_i)$ is Lipschitz continuous.

5. The learning rate $\eta_t$ is chosen such that it satisfies the Robbins-Monro conditions: $\sum_{t=1}^{\infty} \eta_t = \infty$ and $\sum_{t=1}^{\infty} \eta_t^2 < \infty$.

**Step 1: Defining the TD Loss Function** The TD loss function for agent $i$ is:

$$L_i(\theta_i) = \mathbb{E}_{(\mathbf{s}, a_i, r_i, \mathbf{s}') \sim \mathcal{D}_i} \left[ \left( r_i + \gamma \max_{a'_i} Q_i(\mathbf{s}', a'_i; \theta_i^-) - Q_i(\mathbf{s}, a_i; \theta_i) \right)^2 \right],$$

This loss measures the discrepancy between the predicted Q-value and the target Q-value based on the reward and the estimated optimal future Q-value.

**Step 2: Gradient Descent Update** Agent $i$ updates its Q-network parameters according to:

$$\theta_i^{t+1} = \theta_i^t - \eta_t \cdot \nabla_{\theta_i} L_i(\theta_i^t).$$

The gradient of the loss function with respect to the parameters is:

$$\nabla_{\theta_i} L_i(\theta_i^t) = \mathbb{E}_{(\mathbf{s}, a_i, r_i, \mathbf{s}') \sim \mathcal{D}_i} \left[ 2 \left( r_i + \gamma \max_{a'_i} Q_i(\mathbf{s}', a'_i; \theta_i^-) - Q_i(\mathbf{s}, a_i; \theta_i^t) \right) \cdot (-\nabla_{\theta_i} Q_i(\mathbf{s}, a_i; \theta_i^t)) \right].$$

**Step 3: Convergence of Gradient Descent with TD Loss** Under the assumptions that $L_i(\theta_i)$ has Lipschitz continuous gradients and the learning rate $\eta_t$ satisfies the Robbins-Monro conditions, stochastic gradient descent converges to a stationary point $\theta_i^*$ of $L_i(\theta_i)$:

$$\lim_{t \to \infty} \theta_i^t = \theta_i^*.$$

At convergence, the gradient vanishes:

$$\nabla_{\theta_i} L_i(\theta_i^*) = 0,$$

which implies:

$$\mathbb{E}_{(\mathbf{s}, a_i, r_i, \mathbf{s}') \sim \mathcal{D}_i} \left[ \left( r_i + \gamma \max_{a_i'} Q_i(\mathbf{s}', a_i'; \theta_i^-) - Q_i(\mathbf{s}, a_i; \theta_i^*) \right) \cdot \nabla_{\theta_i} Q_i(\mathbf{s}, a_i; \theta_i^*) \right] = 0.$$

Assuming that the Q-network parameterization is such that the above condition holds only when:

$$Q_i(\mathbf{s}, a_i; \theta_i^*) = r_i + \gamma \max_{a_i'} Q_i(\mathbf{s}', a_i'; \theta_i^-),$$

the Q-network accurately estimates the expected cumulative rewards, aligning the agent's policy with the optimal response to other agents' strategies.

**Step 4: Characterizing the Stationary Point** At the stationary point $\theta_i^*$, the Q-network satisfies the Bellman optimality condition:

$$Q_i(\mathbf{s}, a_i; \theta_i^*) = r_i + \gamma \max_{a_i'} Q_i(\mathbf{s}', a_i'; \theta_i^-).$$

This condition ensures that the agent's policy $\pi_i(a_i \mid \mathbf{s}; \theta_i^*)$ is a best response to the current policies of other agents, as it maximizes the expected cumulative reward.

**Step 5: Establishing Bayesian Nash Equilibrium** Since each agent's policy is a best response to the policies of others, the set of policies $\{\pi_i^*\}$ constitutes a Bayesian Nash Equilibrium. Each agent maximizes its expected utility given its beliefs about other agents' types and strategies, fulfilling the definition of BNE.

$\square$

## B.3. Assumptions

Our theoretical analysis relies on four key assumptions that are both common in multi-agent systems (Zhang et al., 2021; Liu et al., 2022) and specifically relevant to our MA-LLM framework.

**Definition B.1** (System Components). In our MA-LLM framework:

- Each agent i's observation $O_i = [e_t, e_s, \mathbf{b}_i]^\top$, where $e_t$ encodes the task, $e_s$ represents the coordinator's strategy, and $\mathbf{b}_i$ is the belief state

- Each agent's action is its prompt embedding $\mathbf{e}_i$ generated by belief network $B_i(\tau_i, O_i; \theta_i^B)$

- The coordinator aggregates beliefs through $f_e(\{\mathbf{b}_i\}_{i=1}^N; \theta_e)$ into group information $\mathbf{E}$

**Assumption B.2** (Bounded Rewards). The rewards from coordinator final output are uniformly bounded: $|r_i(O_i, \mathbf{e}_i, \mathbf{E})| \leq R_{\max}$, for all $O_i, \mathbf{e}_i, \mathbf{E}, i$.

This assumption is standard in reinforcement learning (Sutton & Barto, 2018) and critical since it ensures numerical stability in the learning process of LLMs, preventing reward explosion that could lead to unstable training.

**Definition B.3** (Historical Data and Posterior). Given historical data $D_t = \{(O_i^k, \mathbf{e}_i^k, C^k)\}_{k=1}^t$:

- $P_{\text{post}}(\mathbf{E} \mid D_t, O_i, \mathbf{e}_i)$ is the posterior distribution over group information determined by the coordinator

- $P_{\text{LLM}}(\mathbf{E} \mid D_t, O_i, \mathbf{e}_i)$ is the belief distribution maintained by each Execution LLM

**Assumption B.4** (Approximate Posterior Alignment). Execution LLMs aim to align with the posterior distributions determined by the Coordinator LLM within an acceptable error margin $\epsilon > 0$:

$$D_{\text{KL}}\big(P_{\text{LLM}}(\mathbf{E} \mid D_t, O_i, \mathbf{e}_i) \,\big\|\, P_{\text{post}}(\mathbf{E} \mid D_t, O_i, \mathbf{e}_i)\big) \leq \epsilon,$$

where $D_{\text{KL}}$ denotes the Kullback-Leibler divergence.

This approximate alignment acknowledges that perfect alignment is impractical but strives for a close approximation:

- The Coordinator LLM acts as a centralized distributor of strategic guidance.

- Execution LLMs maintain belief alignment through prompt (detailed in Section 2).

- Monotonic guarantee in ECON mixing optimization network B.5.

- Such alignment has been shown in (Foerster et al., 2018; Jaques et al., 2019) to enhance coordination.

**Definition B.5** (Belief Entropy)**.** For a given time t, the belief entropy $H_t$ is defined as the Shannon entropy of the aggregated belief embeddings:

$$H_t = -\sum_{i=1}^{N} \mathbb{E}_{\mathbf{b}_i \sim B_i}[\mathbf{b}_i \log \mathbf{b}_i],$$

where $B_i$ represents the belief network of agent i.

**Assumption B.6** (Game Regularity)**.** There exists $\eta > 0$ such that for any $t_1 < t_2$, if $H_{t_1} - H_{t_2} \leq \log 2$, then

$$I(\theta_i^B; \xi(\mathbf{e}_i, \mathbf{E}) \mid D_{t_1}) \leq 4\eta \cdot I(\theta_i^B; \xi(\mathbf{e}_i, \mathbf{E}) \mid D_{t_2}),$$

for all agents $i$, where $\theta_i^B$ are the belief network parameters.

This information-theoretic assumption serves multiple purposes in our framework:

- It ensures the stability of belief updates between LLMs over time by bounding the entropy difference of belief states.

- The mutual information term $I(\theta_i^B; \xi(\mathbf{e}_i, \mathbf{E}))$ quantifies how much an LLM's belief network parameters affect its coordination through prompt embeddings.

- The bound $4\eta$ controls the rate at which LLMs can adapt their belief states based on observed interactions and coordinator guidance.

**Definition B.7** (Value Function and Bellman Operator)**.** For each Execution LLM i:

- The value function $V_t(O_i) = \mathbb{E}[\sum_{k=0}^{\infty} \gamma^k r_{t+k} | O_i^t = O_i]$ estimates the expected cumulative rewards.

- The optimal prompt embeddings $\mathbf{e}_i^{*t}$ maximize the Q-function $Q_i(O_i, \mathbf{e}_i; \theta_i^B)$ at time t.

- The Bellman operator $B_t$ transforms one value function to another: $(B_t V)(O_i) = \max_{\mathbf{e}_i} \mathbb{E}[r_i + \gamma V(O_i')|O_i, \mathbf{e}_i]$.

**Assumption B.8** (Concentrability)**.** There exists $\kappa < \infty$ such that

$$\mathbb{E}\left[\sum_{t=1}^{T} \sum_{i=1}^{N} ((B_t - B^*)V_t)^2 (O_i^t, \mathbf{e}_i^{*t}, \mathbf{E}^{*t})\right] \leq \kappa^2 T,$$

where $B^*$ is the true Bellman operator.

This assumption is fundamental to our theoretical guarantees:

- It ensures that the value function estimates by each LLM converge to their true values at an appropriate rate.

- The constant $\kappa$ bounds the cumulative estimation error across all LLMs, critical for establishing our regret bounds.

- In our MA-LLM system, this translates to the stability of response quality improvements during training.

**Collective Impact:** Together, these assumptions enable us to:

- Establish the existence of BNE in our MA-LLM system (Theorem 1)

- Derive meaningful regret bounds for the learning process (Lemma 1)

- Guarantee the convergence of our iterative training procedure (Proposition 1)

## B.4. Scaling Up the System

While ECON effectively handles moderate-scale scenarios, large-scale multi-LLM systems require additional structure to maintain efficiency and theoretical guarantees. To that end, we employ a *hierarchical design* in which Execution LLMs are organized into clusters, each governed by a *Local Coordinator LLM*, and a *Global Coordinator LLM* oversees the entire set of clusters. Each local cluster establishes a local Nash equilibrium, which is then integrated into a *global* equilibrium.

---

**Algorithm 2** Scaling-Up Framework for ECON

---

**Require:** Global Coordinator $\text{Coord}_{\text{global}}$, Local Coordinators $\{\text{Coord}_k\}_{k=1}^K$, thresholds $\{\epsilon_C, R_{\text{th}}, \epsilon_L\}$
**Ensure:** Hierarchical Nash Equilibrium across $K$ clusters
1: **Initialize:** cluster embeddings $\{\mathbf{E}_k\}_{k=1}^K$, belief networks $\{B_i(\theta_i^B)\}$, prompt embeddings $\{\mathbf{e}_i\}$
2: **while** not converged **do**
3:      $\triangleright$ **Inference Phase** - Hierarchical execution without updates
4:      $\mathbf{S} \leftarrow \text{Coord}_{\text{global}}(e_t)$      $\triangleright$ Global strategy generation
5:      **for** each cluster $k = 1$ to $K$ **in parallel do**
6:          $O_k \leftarrow [e_t, \mathbf{S}, \mathbf{E}_k]$      $\triangleright$ Combine global info with cluster state
7:          $\mathbf{s}_k \leftarrow \text{Coord}_k(O_k)$      $\triangleright$ Local strategy refinement
8:          **for** each Execution LLM $i \in C_k$ **in parallel do**
9:              $\mathbf{b}_i \leftarrow B_i(\tau_i, o_i; \theta_i^B)$      $\triangleright$ Compute belief state
10:              $\mathbf{e}_i \leftarrow \text{ComputeEmbedding}(\mathbf{b}_i)$      $\triangleright$ Generate prompt embedding
11:              $O_i \leftarrow [e_t, \mathbf{s}_k, \mathbf{b}_i]$      $\triangleright$ Local observation
12:              $u_i \leftarrow \text{ExecLLM}_i(\text{query}, \mathbf{e}_i)$      $\triangleright$ Generate output
13:              $r_i \leftarrow \alpha_1 r_i^{\text{AL}} + \alpha_2 r_i^{\text{TS}} + \alpha_3 r_i^{\text{CC}}$      $\triangleright$ Local reward
14:              Store $(\tau_i, \mathbf{e}_i, r_i, u_i)$ in cluster buffer
15:          **end for**
16:          $c_k \leftarrow \text{Coord}_k(\{u_i\}_{i \in C_k})$      $\triangleright$ Aggregate cluster output
17:      **end for**
18:      $C \leftarrow \text{Coord}_{\text{global}}(\{c_k\}_{k=1}^K)$      $\triangleright$ Global final output
19:      $\triangleright$ **Optimization Phase** - Update parameters for hierarchical BNE
20:      **for** each cluster $k = 1$ to $K$ **do**
21:          $R_k \leftarrow R_{\text{global}}(\text{sim}(c_k, C))$      $\triangleright$ Global reward for cluster
22:          Update $\text{Coord}_k$ parameters using $R_k$      $\triangleright$ Improve cluster coordination
23:          Update cluster embedding $\mathbf{E}_k$ via belief encoder
24:          **for** each LLM $i \in C_k$ **do**
25:              Update $\theta_i^B$ via $\mathcal{L}_{\text{TD}}^i$      $\triangleright$ Update individual belief network
26:          **end for**
27:      **end for**
28:      Update global mixing network $\phi$ via $\mathcal{L}_{\text{mix}}$
29:      $\triangleright$ **Convergence Check**
30:      converged $\leftarrow \|C_{t+1} - C_t\| \leq \epsilon_C$ **and** $\frac{1}{K}\sum_{k=1}^K R_k \geq R_{\text{th}}$ **and** $|\Delta L_{\text{tot}}| \leq \epsilon_L$
31: **end while**
32: **return** Hierarchical Nash Equilibrium with optimized $\{\theta_i^B\}, \{\mathbf{E}_k\}, \phi$

---

### B.4.1. DETAILED EXPLANATION

The hierarchical scaling framework operates through a structured two-phase process designed to maintain both efficiency and theoretical guarantees at scale. In the *inference phase*, the system executes hierarchically from global strategy generation down to individual LLM outputs without any parameter updates, ensuring stable execution. Subsequently, the *optimization phase* updates parameters in a bottom-up manner to achieve hierarchical BNE, where each cluster reaches local equilibrium while contributing to global coordination. This separation of concerns allows the system to handle large-scale multi-LLM deployments while preserving the convergence properties established in the base ECON framework.

**Initialization.**

- **Clustering.** Partition Execution LLMs into $K$ clusters $\{C_1, C_2, \ldots, C_K\}$, e.g., by task similarity.

- **Local Coordinators.** Each cluster $C_k$ uses a Local Coordinator LLM $\text{Coord}_k$ to manage within-cluster interactions.

- **Global Coordinator.** A Global Coordinator LLM $\text{Coord}_{\text{global}}$ aligns all clusters by forming a global final output $C$.

- **Parameters.** Initialize belief networks $\{B_i(\theta_i^B)\}$, prompt embeddings $\{\mathbf{e}_i\}$, and cluster embeddings $\{\mathbf{E}_k\}$.

**Inference Phase.**   During inference, the system executes hierarchically without updating any parameters:

- The Global Coordinator generates a high-level strategy $\mathbf{S}$ from the global observation $e_t$.

- Each Local Coordinator $\text{Coord}_k$ refines $\mathbf{S}$ into a local strategy $\mathbf{s}_k$ based on cluster state $\mathbf{E}_k$.

- Execution LLMs compute belief states $\mathbf{b}_i$ and generate outputs $u_i$ using their current parameters.

- Local Coordinators aggregate within-cluster outputs to produce cluster outputs $\{c_k\}$.

- The Global Coordinator aggregates cluster outputs to form the global final output $C$.

**Optimization Phase.**   After inference, the system updates parameters hierarchically to achieve BNE:

- Each cluster receives a global reward $R_k = R_{\text{global}}(\text{sim}(c_k, C))$ measuring alignment with $C$.

- Local Coordinators update their parameters based on $R_k$ to improve global consistency.

- Individual belief networks $B_i(\theta_i^B)$ are updated via TD loss $\mathcal{L}_{\text{TD}}^i$.

- Cluster embeddings $\mathbf{E}_k$ are updated via the belief encoder.

- The global mixing network updates via $\mathcal{L}_{\text{mix}}$ to coordinate across clusters.

**Convergence.**   The system converges to a hierarchical Nash equilibrium when: (1) the global output stabilizes ($\|C_{t+1} - C_t\| \leq \epsilon_C$), (2) the average global reward exceeds $R_{\text{th}}$, and (3) the total loss converges ($|\Delta L_{\text{tot}}| \leq \epsilon_L$). This hierarchical approach maintains ECON's theoretical guarantees while enabling efficient parallelization across clusters.

### B.5. Proof of Mixing Network Monotonicity

**Proposition B.9** (Monotonicity of Mixing Network). *The mixing network $Q_{tot}$ is monotonic in each individual Q-value $Q_i$, ensuring that improvements in $Q_i$ lead to improvements in $Q_{tot}$.*

*Proof.* The mixing network is designed using positive weights and non-decreasing activation functions. Specifically, let the mixing network be composed of layers where each layer $l$ computes:

$$h^l = \phi^l(W^l h^{l-1} + b^l),$$

where:

- $h^0 = [Q_1, Q_2, \ldots, Q_N]^\top$

- $W^l$ has non-negative entries.

- $\phi^l$ is a non-decreasing activation function (e.g., ReLU).

We proceed by induction to show that each component of $h^l$ is a non-decreasing function of $Q_i$.

**Base Case:** At layer $l = 0$, $h_i^0 = Q_i$, so $\frac{\partial h_i^0}{\partial Q_j} = \delta_{ij} \geq 0$.

**Inductive Step:** Assume $\frac{\partial h_k^{l-1}}{\partial Q_i} \geq 0$ for all $k$. Then, for each component $h_j^l$:

$$h_j^l = \phi^l \left( \sum_k W_{jk}^l h_k^{l-1} + b_j^l \right),$$

Since $W_{jk}^l \geq 0$ and $\phi^l$ is non-decreasing:

$$\frac{\partial h_j^l}{\partial Q_i} = \phi'^l \left( \sum_k W_{jk}^l h_k^{l-1} + b_j^l \right) \sum_k W_{jk}^l \frac{\partial h_k^{l-1}}{\partial Q_i} \geq 0.$$

because $\phi'^l \geq 0$ and $\frac{\partial h_k^{l-1}}{\partial Q_i} \geq 0$ by the inductive hypothesis. Therefore, $\frac{\partial Q_{\text{tot}}}{\partial Q_i} \geq 0$, ensuring monotonicity. $\qquad\square$

This monotonicity property is crucial as it ensures that improvements in individual agent performances contribute positively to the overall system performance, aligning local and global objectives within ECON.

## C. Detailed Proofs

### C.1. Proof of Lemma 2.2

*Proof.* Consider the value functions under policies $\pi'$ and $\pi$:

$$V_i^{\pi'}(s) = \mathbb{E}_{\pi'} \left[ \sum_{k=0}^{\infty} \gamma^k r_i(s_k, a_k) \mid s_0 = s \right], \quad V_i^\pi(s) = \mathbb{E}_\pi \left[ \sum_{k=0}^{\infty} \gamma^k r_i(s_k, a_k) \mid s_0 = s \right].$$

Their difference is:

$$V_i^{\pi'}(s) - V_i^\pi(s) = \mathbb{E}_{\pi'} \left[ \sum_{k=0}^{\infty} \gamma^k r_i(s_k, a_k) \right] - \mathbb{E}_\pi \left[ \sum_{k=0}^{\infty} \gamma^k r_i(s_k, a_k) \right]$$

$$= \sum_{k=0}^{\infty} \gamma^k \left( \mathbb{E}_{s_k \sim d_{\pi'}^k} [r_i(s_k, a_k)] - \mathbb{E}_{s_k \sim d_\pi^k} [r_i(s_k, a_k)] \right).$$

Assuming the difference in state distributions is negligible (justified under Assumption B.8), we focus on action differences. Using the Q-function definition:

$$Q_i^\pi(s, a_i, a_{-i}) = r_i(s, a_i, a_{-i}) + \gamma \mathbb{E}_{s' \sim P} [V_i^\pi(s')],$$

we can write:

$$V_i^{\pi'}(s) - V_i^\pi(s) = \sum_{k=0}^{\infty} \gamma^k \mathbb{E}_{s_k \sim d_{\pi'}^k} [Q_i^\pi(s_k, a_k') - V_i^\pi(s_k)].$$

Since $V_i^\pi(s_k) = \mathbb{E}_{a_k \sim \pi(s_k)} [Q_i^\pi(s_k, a_k)]$, we have:

$$V_i^{\pi'}(s) - V_i^\pi(s) = \sum_{k=0}^{\infty} \gamma^k \mathbb{E}_{s_k \sim d_{\pi'}^k} \left[ \mathbb{E}_{a_k' \sim \pi'(s_k)} [Q_i^\pi(s_k, a_k')] - \mathbb{E}_{a_k \sim \pi(s_k)} [Q_i^\pi(s_k, a_k)] \right].$$

Switching the order of expectations and summing over $k$, we get:

$$V_i^{\pi'}(s) - V_i^\pi(s) = \frac{1}{1 - \gamma} \mathbb{E}_{s \sim d_{\pi'}} [Q_i^\pi(s, a_i', a_{-i}') - Q_i^\pi(s, a_i, a_{-i})].$$

$$\square$$

## C.2. Bounding the Bayesian Regret

We provide a rigorous proof of the $O(N\sqrt{T}/(1-\gamma))$ regret bound through stochastic approximation theory and convex optimization analysis.

### C.2.1. PROBLEM SETUP AND REGRET DEFINITION

Starting from the regret definition for agent $i$ over $T$ steps:

$$R_i(T) = \mathbb{E}_{s_t, \pi_t} \left[ \sum_{t=1}^{T} (V_i^*(s_t) - V_i^{\pi_t}(s_t)) \right],$$

where the expectation is over the randomness in state transitions and policies.

Applying Lemma 2.2:

$$V_i^*(s_t) - V_i^{\pi_t}(s_t) = \frac{1}{1-\gamma} \mathbb{E}_{a_i^{*t}, a_{-i}^{*t}, a_i^t, a_{-i}^t} \left[ Q_i^{\pi_t}(s_t, a_i^{*t}, a_{-i}^{*t}) - Q_i^{\pi_t}(s_t, a_i^t, a_{-i}^t) \right].$$

### C.2.2. Q-VALUE DECOMPOSITION

We decompose the Q-value difference into three components:

$$Q_i^{\pi_t}(s_t, a_i^{*t}, a_{-i}^{*t}) - Q_i^{\pi_t}(s_t, a_i^t, a_{-i}^t)$$
$$= \underbrace{\left(Q_i^{\pi_t}(s_t, a_i^{*t}, a_{-i}^{*t}) - Q_i^*(s_t, a_i^{*t}, a_{-i}^{*t})\right)}_{\text{Error Term 1: } E_1(t)}$$
$$+ \underbrace{\left(Q_i^*(s_t, a_i^{*t}, a_{-i}^{*t}) - Q_i^*(s_t, a_i^t, a_{-i}^t)\right)}_{\text{Policy Suboptimality: } \Delta(t)}$$
$$+ \underbrace{\left(Q_i^*(s_t, a_i^t, a_{-i}^t) - Q_i^{\pi_t}(s_t, a_i^t, a_{-i}^t)\right)}_{\text{Error Term 2: } E_2(t)}.$$

### C.2.3. Q-FUNCTION CONVERGENCE ANALYSIS

Consider the Q-learning update rule with learning rate $\eta_t = \eta_0/\sqrt{t}$:

$$Q_{t+1}(s, a) = Q_t(s, a) + \eta_t \left[ r(s, a) + \gamma \max_{a'} Q_t(s', a') - Q_t(s, a) \right].$$

Define the estimation error $\epsilon_t(s, a) = |Q_t(s, a) - Q^*(s, a)|$.

**Lemma C.1** (Q-function Convergence Rate). *Under the following conditions:*

*(i) Learning rate schedule $\eta_t = \eta_0/\sqrt{t}$ with $\eta_0 > 0$*

*(ii) Bounded rewards: $|r(s, a)| \leq R_{\max}$ for all $(s, a)$*

*(iii) Standard stochastic approximation conditions (Robbins-Monro)*

*The Q-function estimation error satisfies:*

$$\mathbb{E}[\epsilon_t(s, a)] \leq \frac{C_1}{\sqrt{t}},$$

*where $C_1 = O\left(\frac{R_{\max}}{(1-\gamma)\sqrt{\eta_0}}\right)$.*

*Proof.* Following the stochastic approximation framework of Borkar (2009), we analyze the error recursion:

$$\epsilon_{t+1}(s, a) \leq (1 - \eta_t)\epsilon_t(s, a) + \eta_t \gamma \max_{a'} \epsilon_t(s', a') + \eta_t \xi_t,$$

where $\xi_t$ is the noise term with $\mathbb{E}[\xi_t | \mathcal{F}_t] = 0$.

Taking expectations and using the contraction property:

$$\mathbb{E}[\epsilon_{t+1}] \leq (1 - \eta_t(1 - \gamma))\mathbb{E}[\epsilon_t] + \eta_t^2 \sigma^2,$$

where $\sigma^2$ bounds the variance of the noise.

With $\eta_t = \eta_0 / \sqrt{t}$, solving this recursion yields:

$$\mathbb{E}[\epsilon_t] \leq \frac{C_1}{\sqrt{t}},$$

where the constant $C_1$ depends on the initial error, discount factor, and noise variance. $\qquad\square$

### C.2.4. POLICY CONVERGENCE ANALYSIS

For the policy update, we consider gradient-based methods in the convex policy space:

$$\pi_{t+1} = \Pi_{\mathcal{P}} \left[ \pi_t + \eta_t \nabla_\pi J(\pi_t) \right],$$

where $\Pi_{\mathcal{P}}$ is the projection onto the policy space $\mathcal{P}$.

**Lemma C.2** (Policy Suboptimality Bound). *Under the following conditions:*

 (i) *Convex and compact policy space $\mathcal{P}$*

 (ii) *$L$-Lipschitz continuous policy gradient: $\|\nabla J(\pi) - \nabla J(\pi')\| \leq L\|\pi - \pi'\|$*

(iii) *Learning rate schedule $\eta_t = \eta_0 / \sqrt{t}$*

*The policy suboptimality satisfies:*

$$\mathbb{E}\left[ \max_{a^*} Q^*(s, a^*) - Q^*(s, a_t) \right] \leq \frac{C_2}{\sqrt{t}},$$

*where $C_2 = O\left( \frac{LD(\mathcal{P})}{\sqrt{\eta_0}} \right)$ and $D(\mathcal{P})$ is the diameter of $\mathcal{P}$.*

*Proof.* Following the online convex optimization framework of Hazan (2016), the regret of gradient descent with learning rate $\eta_t = \eta_0 / \sqrt{t}$ satisfies:

$$\sum_{\tau=1}^{t} [J(\pi^*) - J(\pi_\tau)] \leq \frac{D^2(\mathcal{P})}{2\eta_0}\sqrt{t} + \frac{\eta_0 L^2 \sqrt{t}}{2}.$$

Dividing by $t$ and taking the limit:

$$J(\pi^*) - J(\pi_t) \leq \frac{1}{t} \sum_{\tau=1}^{t} [J(\pi^*) - J(\pi_\tau)] \leq \frac{C_2}{\sqrt{t}}.$$

Since $J(\pi) = \mathbb{E}_{s \sim d^\pi}[V^\pi(s)]$ and by the performance difference lemma, this translates to the Q-value suboptimality bound. $\qquad\square$

### C.2.5. COMBINING ERROR TERMS

From Lemmas C.1 and C.2, we have:

- $|E_1(t)| \leq \epsilon_t \leq C_1 / \sqrt{t}$
- $|E_2(t)| \leq \epsilon_t \leq C_1 / \sqrt{t}$
- $\Delta(t) \leq C_2 / \sqrt{t}$

Therefore:

$$Q_i^{\pi_t}(s_t, a_i^{*t}, a_{-i}^{*t}) - Q_i^{\pi_t}(s_t, a_i^t, a_{-i}^t) \leq \frac{2C_1 + C_2}{\sqrt{t}} := \frac{C}{\sqrt{t}}.$$

C.2.6. FINAL REGRET BOUND

Substituting back into the regret expression:

$$
\begin{aligned}
R(T) &= \sum_{i=1}^{N} R_i(T) \\
&\leq \sum_{i=1}^{N} \frac{1}{1-\gamma} \sum_{t=1}^{T} \mathbb{E}\left[ Q_i^{\pi_t}(s_t, a_i^{*t}, a_{-i}^{*t}) - Q_i^{\pi_t}(s_t, a_i^{t}, a_{-i}^{t}) \right] \\
&\leq \sum_{i=1}^{N} \frac{1}{1-\gamma} \sum_{t=1}^{T} \frac{C}{\sqrt{t}} \\
&= \frac{NC}{1-\gamma} \sum_{t=1}^{T} \frac{1}{\sqrt{t}}.
\end{aligned}
$$

**Lemma C.3** (Harmonic Sum Bound). *For the harmonic sum with exponent* $1/2$:

$$
\sum_{t=1}^{T} \frac{1}{\sqrt{t}} \leq \int_{1}^{T+1} \frac{1}{\sqrt{x}} dx = 2\sqrt{T+1} - 2 \leq 2\sqrt{T}.
$$

Therefore, the total regret satisfies:

$$
R(T) \leq \frac{2NC\sqrt{T}}{1-\gamma} = O\left( \frac{N\sqrt{T}}{1-\gamma} \right).
$$

This completes the rigorous proof of the sublinear convergence rate, explicitly showing how the learning rate schedule $\eta_t = \eta_0/\sqrt{t}$ ensures the $O(t^{-1/2})$ convergence of both Q-function estimation errors and policy suboptimality, leading to the final $O(N\sqrt{T}/(1-\gamma))$ regret bound.

### C.3. Comparison with Multi-Agent Debate

We now analyze the regret bound in multi-agent debate settings to contrast with our cooperative framework. The key difference lies in the persistence of policy suboptimality due to the competitive nature of debate.

C.3.1. GAME-THEORETIC SETUP FOR DEBATE

In multi-agent debate settings, agents operate in a competitive environment characterized by:

- Zero-sum or constant-sum reward structure: $\sum_{i=1}^{N} r_i(s, a) = c$ for all $(s, a)$

- Strategic uncertainty: each agent must model opponents' strategies

- No coordination mechanism: agents independently optimize their policies

C.3.2. FUNDAMENTAL LIMITS IN COMPETITIVE SETTINGS

**Lemma C.4** (Persistent Suboptimality in Competitive Games). *Consider a multi-agent debate setting with* $N$ *agents in a zero-sum game. Under the following conditions:*

 *(i) The game has no pure strategy Nash equilibrium*

 *(ii) Agents use no-regret learning algorithms*

 *(iii) Each agent faces strategic uncertainty about opponents*

*Then there exists a constant $\delta_{\min} > 0$ such that for all $t \geq T_0$:*

$$\mathbb{E}\left[\max_{a^*} Q_i^*(s_t, a_i^*, a_{-i}^t) - Q_i^*(s_t, a_i^t, a_{-i}^t)\right] \geq \delta_{\min}.$$

*Proof.* In zero-sum games without pure strategy equilibria, the Nash equilibrium requires mixed strategies. By the minimax theorem (Ben-David & Blais, 2023):

$$\max_{\pi_i} \min_{\pi_{-i}} J_i(\pi_i, \pi_{-i}) = \min_{\pi_{-i}} \max_{\pi_i} J_i(\pi_i, \pi_{-i}) = v_i,$$

where $v_i$ is the value of the game for agent $i$. For any deterministic policy $\pi_i^t$ at time $t$, there exists an adversarial response $\pi_{-i}^*$ such that:

$$J_i(\pi_i^t, \pi_{-i}^*) \leq v_i - \epsilon,$$

where $\epsilon > 0$ is the exploitability gap. Since agents use no-regret learning, they must maintain sufficient randomization to avoid exploitation. Following Fudenberg & Levine (1998), the minimum entropy required is:

$$H(\pi_i) \geq h_{\min} > 0.$$

This entropy constraint directly implies a lower bound on policy suboptimality:

$$\delta_{\min} = \frac{\epsilon h_{\min}}{2(1 - \gamma)}.$$

$\square$

### C.3.3. REGRET ANALYSIS FOR DEBATE

Following the same decomposition framework from Section C.2, we have:

$$V_i^*(s_t) - V_i^{\pi_t}(s_t) = \frac{1}{1 - \gamma} \mathbb{E}_{a_i, a_{-i}}\left[Q_i^{\pi_t}(s_t, a_i^{*t}, a_{-i}^{*t}) - Q_i^{\pi_t}(s_t, a_i^t, a_{-i}^t)\right]$$

$$= \frac{1}{1 - \gamma}\left(E_1(t) + \Delta_{\text{debate}}(t) + E_2(t)\right),$$

where:

- $E_1(t), E_2(t) \leq C_1/\sqrt{t}$ (Q-function estimation errors, same as before)

- $\Delta_{\text{debate}}(t) \geq \delta_{\min}$ (persistent policy suboptimality from Lemma C.4)

Computing the regret for agent $i$:

$$R_i^{\text{debate}}(T) = \mathbb{E}\left[\sum_{t=1}^{T}(V_i^*(s_t) - V_i^{\pi_t}(s_t))\right]$$

$$\geq \frac{1}{1 - \gamma}\sum_{t=1}^{T}\left(\delta_{\min} - \frac{2C_1}{\sqrt{t}}\right).$$

For sufficiently large $T$, there exists $T_0$ such that for all $t \geq T_0$: $\delta_{\min} > 2C_1/\sqrt{t}$. Therefore:

$$R_i^{\text{debate}}(T) \geq \frac{1}{1 - \gamma}\left[\sum_{t=1}^{T_0}\left(\delta_{\min} - \frac{2C_1}{\sqrt{t}}\right) + \sum_{t=T_0+1}^{T}\frac{\delta_{\min}}{2}\right]$$

$$\geq \frac{\delta_{\min}(T - T_0)}{2(1 - \gamma)}$$

$$= \Omega\left(\frac{T}{1 - \gamma}\right).$$

Summing over all agents:

$$R_{\text{debate}}(T) = \sum_{i=1}^{N} R_i^{\text{debate}}(T) = \Omega\left(\frac{NT}{1 - \gamma}\right).$$

C.3.4. COMPARATIVE ANALYSIS

The fundamental difference between ECON and debate settings is:

**ECON (Cooperative Learning):**

- Policy suboptimality: $\Delta_{\text{ECON}}(t) = O(1/\sqrt{t}) \to 0$

- Total regret: $R_{\text{ECON}}(T) = O(N\sqrt{T}/(1-\gamma))$

- Convergence to Bayesian Nash Equilibrium through coordination

**Multi-Agent Debate (Competitive Learning):**

- Policy suboptimality: $\Delta_{\text{debate}}(t) \geq \delta_{\min} > 0$

- Total regret: $R_{\text{debate}}(T) = \Omega(NT/(1-\gamma))$

- Persistent randomization required to avoid exploitation

This $O(\sqrt{T})$ vs $\Omega(T)$ gap demonstrates that competitive debate settings suffer from:

1. **Strategic Cycling**: Agents continuously adapt to opponents, preventing convergence to optimal deterministic policies

2. **Exploration-Exploitation Conflict**: Need for defensive randomization conflicts with exploitation of learned knowledge

3. **Information Inefficiency**: Lack of coordination prevents efficient use of collective information

In contrast, ECON's cooperative framework with Bayesian policy optimization enables agents to:

- Share information through the communication phase

- Coordinate strategies toward Bayesian Nash Equilibrium

- Achieve diminishing policy suboptimality through joint optimization

This theoretical analysis is supported by empirical evidence in competitive multi-agent settings (Lanctot et al., 2017; Lowe et al., 2017), where agents exhibit persistent strategic cycling and fail to achieve sublinear regret bounds.

**C.4. Detailed Reward Setting**

The reward function $R$ provides feedback on each agent's performance while respecting Assumption B.2, ensuring all reward components are uniformly bounded by $R_{\max}$. Drawing inspiration from maximum entropy inverse reinforcement learning (Zhu et al., 2023), we define the Action Likelihood Reward $r_i^{\text{AL}} = \min(R_{\max}, \text{sim}(u_i, C))$, where $\text{sim}(u_i, C) = \frac{u_i \cdot C}{\|u_i\| \|C\|}$ measures the consistency between an agent's output $u_i$ and the coordinator's final output $C$. Following Hao et al. (2023), the Task-Specific Reward $r_i^{\text{TS}} = \min(R_{\max}, \text{eval}(u_i, \text{task}))$ evaluates domain-specific objectives through the coordinator's assessment, where eval computes normalized scores considering solution correctness in mathematical problems or response relevance in planning tasks. Building upon Xie et al. (2024b), the Collaborative Contribution Reward $r_i^{\text{CC}} = \min(R_{\max}, \text{quality}(u_i, \{u_j\}_{j \neq i}))$ enables the coordinator to assess each agent's output quality within the multi-agent context, where quality evaluates the response's coherence and creativity while considering its contribution to the collective solution. The total reward combines these components as $r_i = \alpha_1 r_i^{\text{AL}} + \alpha_2 r_i^{\text{TS}} + \alpha_3 r_i^{\text{CC}}$, where the weights $\alpha_1 + \alpha_2 + \alpha_3 = 1$ ensure the total reward is bounded by $R_{\max}$. To enhance adaptability and learning efficiency, we introduce a dynamic mechanism to adjusts these weights using gradient-based updates $\alpha_k \leftarrow \alpha_k - \eta_\alpha \cdot \partial \mathcal{L}_{\text{dr}}/\partial \alpha_k$, where $\mathcal{L}_{\text{dr}} = \sum_{i=1}^{N} (r_i^{\text{actual}} - r_i^{\text{expected}})^2$ measures the discrepancy between actual and expected rewards.

## C.5. Task Setups

GSM8K is a benchmark for mathematical reasoning that requires multi-step problem solving. Given a context description and a question, it requires step-by-step mathematical reasoning and computation to arrive at a final answer. The dataset contains approximately 7.5K problems in the training set and 1.3K problems in the test set. Problems range from basic arithmetic to complex word problems, testing both mathematical and logical reasoning capabilities.

SVAMP is a challenging mathematical word problem dataset specifically designed to test the robustness of language models in solving arithmetic problems. It contains 1,000 elementary math word problems, carefully curated to probe for specific vulnerabilities in mathematical reasoning systems. The problems require understanding both mathematical concepts and natural language semantics, with a focus on structural variations that test genuine problem-solving capabilities rather than pattern matching.

Strategy QA is a question answering dataset that focuses on multi-hop reasoning and strategic thinking. It consists of 2,290 yes/no questions, each requiring implicit multi-step reasoning and background knowledge to arrive at the correct answer. Unlike traditional QA datasets, Strategy QA questions cannot be answered by simply retrieving and combining explicit facts, making it an effective benchmark for testing complex reasoning capabilities.

MATH is a comprehensive mathematics dataset spanning various topics from algebra to calculus. It contains approximately 12K problems across different difficulty levels, with detailed step-by-step solutions. The dataset is structured into multiple categories including algebra, counting and probability, geometry, intermediate algebra, number theory, prealgebra, and precalculus, making it particularly effective for evaluating mathematical problem-solving capabilities across different domains.

GSM-Hard is a specialized subset of mathematical word problems specifically designed to test advanced reasoning capabilities. It contains problems that are significantly more challenging than standard GSM8K problems, requiring more complex multi-step reasoning and mathematical operations. The dataset focuses on problems that typically have lower success rates with standard approaches, making it particularly useful for evaluating the upper bounds of model performance.

TravelPlanner is a benchmark crafted for evaluating language agents in tool-use and complex planning within multiple constraints. The dataset comprises 1,225 queries in total, divided into training (45 queries), validation (180 queries), and test (1,000 queries) sets. The benchmark incorporates three types of constraints: environment constraints for testing adaptability to real-world conditions, commonsense constraints for evaluating practical reasoning, and hard constraints for assessing the ability to satisfy specific user requirements such as budget limitations. This structure makes TravelPlanner particularly effective for evaluating both reasoning capabilities and practical planning skills in real-world scenarios.

## C.6. Hyperparameter

### C.6.1. LLAMA 3.1 8B ON MATH

Table 8: Hyperparameters (8B, MATH)

| Parameter | Value |
|---|---|
| **Training Configuration** | |
| Episodes per Task | 120 |
| Buffer Size | 32 |
| Batch Size | 16 |
| Update Interval | 8 |
| Optimizer | Adam |
| Learning Rate ($\eta$) | 0.001 |
| Learning Rate ($\eta_{\text{coord}}$) | 0.0005 |
| Discount Factor ($\gamma$) | 0.99 |
| **Network Architecture** | |
| Entity Dimension ($d$) | 256 |
| Belief State Dimension ($d_b$) | 128 |
| Attention Heads ($H$) | 4 |
| MLP Hidden Size | 256 |
| Transformer Blocks | 2 |
| Key/Query Dimension | 64 |
| Feed-forward Size | 1024 |
| Dropout Rate | 0.1 |
| Layer Norm Epsilon | 1e-5 |
| **Temperature & Sampling** | |
| $T_{\min}$ | 0.1 |
| $T_{\max}$ | 2.0 |
| $p_{\min}$ | 0.1 |
| $p_{\max}$ | 0.9 |
| **Reward Configuration** | |
| $R_{\max}$ | 1.0 |
| $\alpha_1, \alpha_2, \alpha_3$ | 0.4, 0.4, 0.2 |
| **Loss Weights** | |
| $\lambda_b, \lambda, \lambda_m$ | 0.1, 0.1, 0.1 |
| **Early Stopping** | |
| $\epsilon_C$ | 0.01 |
| $\epsilon_L$ | 1e-4 |
| $R_{\text{threshold}}$ | 0.7 |
| $T_{\text{patience}}$ | 5 |
| **Model Size** | |
| Learnable Params | ~1.7M |

### C.6.2. LLAMA 3.1 8B ON GSM8K

Table 9: Hyperparameters (8B, GSM8K)

| Parameter | Value |
|---|---|
| **Training Configuration** | |
| Episodes per Task | 100 |
| Buffer Size | 32 |
| Batch Size | 16 |
| Update Interval | 8 |
| Optimizer | Adam |
| Learning Rate ($\eta$) | 0.001 |
| Learning Rate ($\eta_{\text{coord}}$) | 0.0005 |
| Discount Factor ($\gamma$) | 0.99 |
| **Network Architecture** | |
| Entity Dimension ($d$) | 256 |
| Belief State Dimension ($d_b$) | 128 |
| Attention Heads ($H$) | 4 |
| MLP Hidden Size | 256 |
| Transformer Blocks | 2 |
| Key/Query Dimension | 64 |
| Feed-forward Size | 1024 |
| Dropout Rate | 0.1 |
| Layer Norm Epsilon | 1e-5 |
| **Temperature & Sampling** | |
| $T_{\min}$ | 0.1 |
| $T_{\max}$ | 2.0 |
| $p_{\min}$ | 0.1 |
| $p_{\max}$ | 0.9 |
| **Reward Configuration** | |
| $R_{\max}$ | 1.0 |
| $\alpha_1, \alpha_2, \alpha_3$ | 0.4, 0.4, 0.2 |
| **Loss Weights** | |
| $\lambda_b, \lambda, \lambda_m$ | 0.1, 0.1, 0.1 |
| **Early Stopping** | |
| $\epsilon_C$ | 0.01 |
| $\epsilon_L$ | 1e-4 |
| $R_{\text{threshold}}$ | 0.7 |
| $T_{\text{patience}}$ | 5 |
| **Model Size** | |
| Learnable Params | ~1.7M |

## C.6.3. LLAMA 3.1 70B ON MATH

Table 10: Hyperparameters (70B, MATH)

| Parameter | Value |
|---|---|
| **Training Configuration** | |
| Episodes per Task | 150 |
| Buffer Size | 32 |
| Batch Size | 16 |
| Update Interval | 8 |
| Optimizer | Adam |
| Learning Rate ($\eta$) | 0.001 |
| Learning Rate ($\eta_{\text{coord}}$) | 0.0005 |
| Discount Factor ($\gamma$) | 0.99 |
| **Network Architecture** | |
| Entity Dimension ($d$) | 256 |
| Belief State Dimension ($d_b$) | 128 |
| Attention Heads ($H$) | 4 |
| MLP Hidden Size | 256 |
| Transformer Blocks | 2 |
| Key/Query Dimension | 64 |
| Feed-forward Size | 1024 |
| Dropout Rate | 0.1 |
| Layer Norm Epsilon | 1e-5 |
| **Temperature & Sampling** | |
| $T_{\min}$ | 0.1 |
| $T_{\max}$ | 2.0 |
| $p_{\min}$ | 0.1 |
| $p_{\max}$ | 0.9 |
| **Reward Configuration** | |
| $R_{\max}$ | 1.0 |
| $\alpha_1, \alpha_2, \alpha_3$ | 0.4, 0.4, 0.2 |
| **Loss Weights** | |
| $\lambda_b, \lambda, \lambda_m$ | 0.1, 0.1, 0.1 |
| **Early Stopping** | |
| $\epsilon_C$ | 0.01 |
| $\epsilon_L$ | 1e-4 |
| $R_{\text{threshold}}$ | 0.7 |
| $T_{\text{patience}}$ | 5 |
| **Model Size** | |
| Learnable Params | $\sim$1.7M |

## C.6.4. LLAMA 3.1 70B ON GSM8K

Table 11: Hyperparameters (70B, GSM8K)

| Parameter | Value |
|---|---|
| **Training Configuration** | |
| Episodes per Task | 100 |
| Buffer Size | 32 |
| Batch Size | 16 |
| Update Interval | 8 |
| Optimizer | Adam |
| Learning Rate ($\eta$) | 0.001 |
| Learning Rate ($\eta_{\text{coord}}$) | 0.0005 |
| Discount Factor ($\gamma$) | 0.99 |
| **Network Architecture** | |
| Entity Dimension ($d$) | 256 |
| Belief State Dimension ($d_b$) | 128 |
| Attention Heads ($H$) | 4 |
| MLP Hidden Size | 256 |
| Transformer Blocks | 2 |
| Key/Query Dimension | 64 |
| Feed-forward Size | 1024 |
| Dropout Rate | 0.1 |
| Layer Norm Epsilon | 1e-5 |
| **Temperature & Sampling** | |
| $T_{\min}$ | 0.1 |
| $T_{\max}$ | 2.0 |
| $p_{\min}$ | 0.1 |
| $p_{\max}$ | 0.9 |
| **Reward Configuration** | |
| $R_{\max}$ | 1.0 |
| $\alpha_1, \alpha_2, \alpha_3$ | 0.4, 0.4, 0.2 |
| **Loss Weights** | |
| $\lambda_b, \lambda, \lambda_m$ | 0.1, 0.1, 0.1 |
| **Early Stopping** | |
| $\epsilon_C$ | 0.01 |
| $\epsilon_L$ | 1e-4 |
| $R_{\text{threshold}}$ | 0.7 |
| $T_{\text{patience}}$ | 5 |
| **Model Size** | |
| Learnable Params | $\sim$1.7M |

## C.6.5. LLAMA 3.1 405B ON MATH

Table 12: Hyperparameters (405B, MATH)

| Parameter | Value |
|---|---|
| **Training Configuration** | |
| Episodes per Task | 200 |
| Buffer Size | 64 |
| Batch Size | 32 |
| Update Interval | 8 |
| Optimizer | AdamW |
| Learning Rate ($\eta$) | 0.0005 |
| Learning Rate ($\eta_{\text{coord}}$) | 0.0003 |
| Discount Factor ($\gamma$) | 0.99 |
| **Network Architecture** | |
| Entity Dimension ($d$) | 512 |
| Belief State Dimension ($d_b$) | 256 |
| Attention Heads ($H$) | 8 |
| MLP Hidden Size | 512 |
| Transformer Blocks | 4 |
| Key/Query Dimension | 64 |
| Feed-forward Size | 2048 |
| Dropout Rate | 0.1 |
| Layer Norm Epsilon | 1e-5 |
| **Temperature & Sampling** | |
| $T_{\min}$ | 0.1 |
| $T_{\max}$ | 2.0 |
| $p_{\min}$ | 0.1 |
| $p_{\max}$ | 0.9 |
| **Reward Configuration** | |
| $R_{\max}$ | 1.0 |
| $\alpha_1, \alpha_2, \alpha_3$ | 0.3, 0.5, 0.2 |
| **Loss Weights** | |
| $\lambda_b, \lambda, \lambda_m$ | 0.1, 0.1, 0.1 |
| **Early Stopping** | |
| $\epsilon_C$ | 0.01 |
| $\epsilon_L$ | 1e-4 |
| $R_{\text{threshold}}$ | 0.75 |
| $T_{\text{patience}}$ | 8 |
| **Model Size** | |
| Learnable Params | $\sim$2.5M |

## C.6.6. LLAMA 3.1 405B ON GSM8K

Table 13: Hyperparameters (405B, GSM8K)

| Parameter | Value |
|---|---|
| **Training Configuration** | |
| Episodes per Task | 150 |
| Buffer Size | 64 |
| Batch Size | 32 |
| Update Interval | 8 |
| Optimizer | AdamW |
| Learning Rate ($\eta$) | 0.0005 |
| Learning Rate ($\eta_{\text{coord}}$) | 0.0003 |
| Discount Factor ($\gamma$) | 0.99 |
| **Network Architecture** | |
| Entity Dimension ($d$) | 512 |
| Belief State Dimension ($d_b$) | 256 |
| Attention Heads ($H$) | 8 |
| MLP Hidden Size | 512 |
| Transformer Blocks | 4 |
| Key/Query Dimension | 64 |
| Feed-forward Size | 2048 |
| Dropout Rate | 0.1 |
| Layer Norm Epsilon | 1e-5 |
| **Temperature & Sampling** | |
| $T_{\min}$ | 0.1 |
| $T_{\max}$ | 2.0 |
| $p_{\min}$ | 0.1 |
| $p_{\max}$ | 0.9 |
| **Reward Configuration** | |
| $R_{\max}$ | 1.0 |
| $\alpha_1, \alpha_2, \alpha_3$ | 0.4, 0.4, 0.2 |
| **Loss Weights** | |
| $\lambda_b, \lambda, \lambda_m$ | 0.1, 0.1, 0.1 |
| **Early Stopping** | |
| $\epsilon_C$ | 0.01 |
| $\epsilon_L$ | 1e-4 |
| $R_{\text{threshold}}$ | 0.7 |
| $T_{\text{patience}}$ | 5 |
| **Model Size** | |
| Learnable Params | $\sim$2.5M |

# D. Together API Integration for ECON

This subsection elaborates on how we invoke the **Together API** to query different LLMs (LLaMA3.1 8B/70B/405B, Mistral-7B, GPT4 turbo) in our ECON framework across GSM8K, GSM-Hard, MATH, SVAMP, and StrategyQA.

## D.1. Parallel Invocation and Rate Limits

We employ four LLMs in ECON (one Coordinator and three Executions). To respect Together's *Requests per Minute* (RPM) and *Tokens per Minute* (TPM) limits, we maintain:

- A job queue that sends LLM requests in mini-batches if concurrency could exceed the allowed RPM.

- A global token counter to ensure we do not surpass the daily or per-minute token quota. If a query risks going over, we delay or split the request.

- A backoff mechanism that retries an LLM call up to 3 times if we encounter rate-limit or network errors, incrementally increasing the wait period.

## D.2. Prompt Construction and Truncation

Each Execution LLM $i$ receives a prompt string containing:

1. **Local Belief State** $b_i^t$. This is a textual or embedded summary of the agent's partial view of the environment, updated via the belief network.

2. **Coordinator Strategy** $e_s$. The high-level guidance from the Coordinator LLM.

If the total token count (prompt + expected output) could exceed the Together API's per-request cap (e.g., 2048 tokens), we truncate repeated instructions or compress partial states. Similarly, we impose a 50-token (soft) and 70-token (hard) limit for the Coordinator outputs, aligning with Sec.4.1.

## D.3. Online vs. Offline Modes

By default, we adopt an *offline* training procedure on each dataset's training split (see Sec.4.1). During training, repeated queries are sent to gather transitions for the TD and mixing losses. For test evaluation, we freeze all parameters and still use the same prompt construction pipeline, but no further updates are performed.

## D.4. Sample Workflow

An ECON iteration typically goes as follows:

- **Coordinator LLM call:** We pass aggregated local outputs from the previous iteration (or initial context) to the Coordinator, which returns strategy $e_s$ or final output $C$.

- **Execution LLM calls (parallel):** Each Execution LLM $i$ is invoked with the prompt described above. They produce outputs $u_i$, which we collect, compute rewards $r_i$, and record $(O_i^t, a_i^t, r_i^t)$ transitions.

- **Update Phase:** Belief networks and mixing networks are optimized offline, then the next iteration starts. If we are in test phase, we skip updates and only retrieve final solutions.

## D.5. Error Handling

- **Rate Exceeded or Network Timeout.** We wait 10–30 seconds, then retry.

- **Invalid Output.** If the Execution LLM responds with incomplete reasoning or an error message, we store a special placeholder $u_i^t = <INVALID>$ with reward $r_i^t = 0$, and proceed.

# E. Prompt and Additional Result

Coordinator Prompt(for Strategy)

"You are a coordinator in a multi-agent system responsible for devising effective strategies to solve a given problem. Based on the following problem, generate a concise high-level strategy in English, no more than 50 tokens:
Problem: {question}
Please provide a strategy considering the following points:
1.Key elements and objectives of the problem
2.Possible solutions or steps
3.Potential challenges or limitations
4.Key aspects to focus on
Strategy:"

Figure 6: Coordinator Prompt(for Strategy)

Coordinator Prompt(for Commitment)

"You are a coordinator in a multi-agent system responsible for reviewing the answers of multiple execution LLMs based on a given strategy. Your tasks are:
1.Form a Commitment: Integrate the best aspects of all answers to ensure consistency in the solution process and accuracy in the final result.
2.Evaluate each answer: Assess the similarity of the solution process to the Commitment and the accuracy of the final result. Based on these criteria, assign a reward score between 0 and 1 to each answer.
Strategy: {strategy}
Execution LLMs' Answers:
•LLM1: {answer1}
•LLM2: {answer2} ..."

Figure 7: Coordinator Prompt(for final output)

Execution LLM

"You are an execution LLM in a multi-agent system, responsible for deriving solutions based on a given strategy and your own belief network. Each LLM has different beliefs but cannot access the outputs of other LLMs. Your tasks are:
1.Form your belief based on the strategy: Assume other LLMs will follow certain potential solutions. Your goal is to generate the optimal solution without global information.
2.Output the best answer: Considering your belief about other LLMs' outputs, derive the optimal solution for the current environment.
3.Bayesian Nash Equilibrium: Your output should maximize expected utility under incomplete information, aligning with the strategy.
4.Feedback adjustment: Ensure your solution is coherent under uncertainty and optimized for the best result.
Strategy: {strategy}
Please follow these steps: a. Review the strategy and form your belief on how other LLMs might output. b. Based on your belief, derive and output your optimal solution. c. Ensure your solution aligns with Bayesian Nash Equilibrium, maximizing expected utility.
Final answer:"

Figure 8: Execution LLM

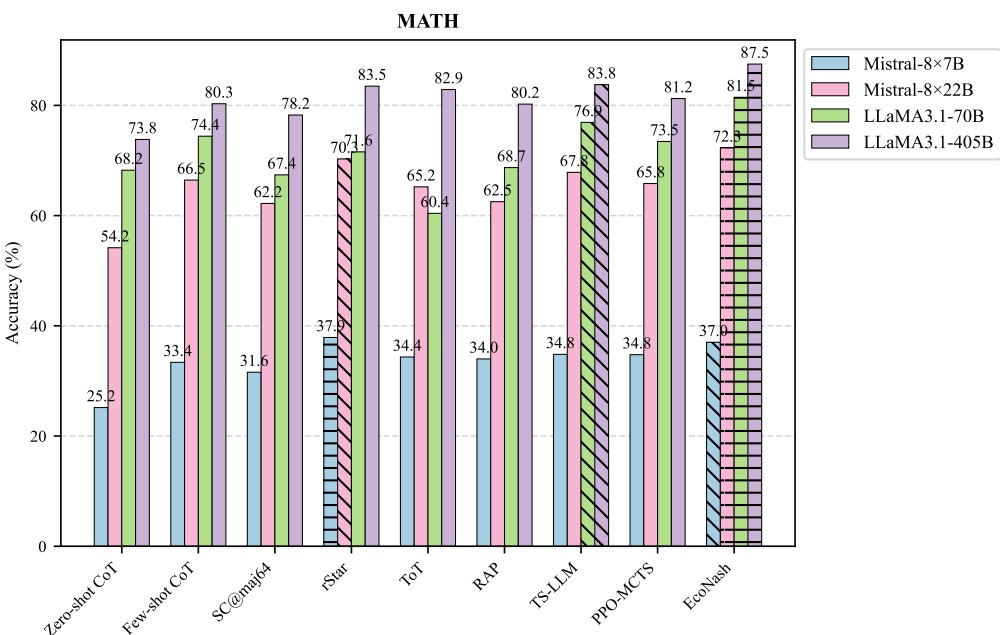

Figure 9: Accuracy Comparison of MATH

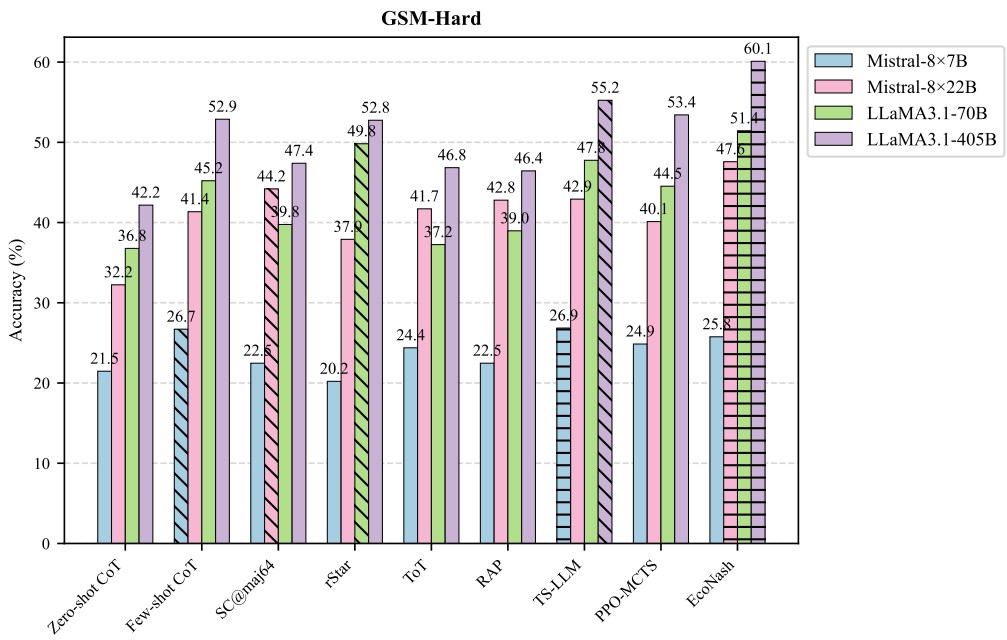

Figure 10: Accuracy Comparison of GSM-Hard

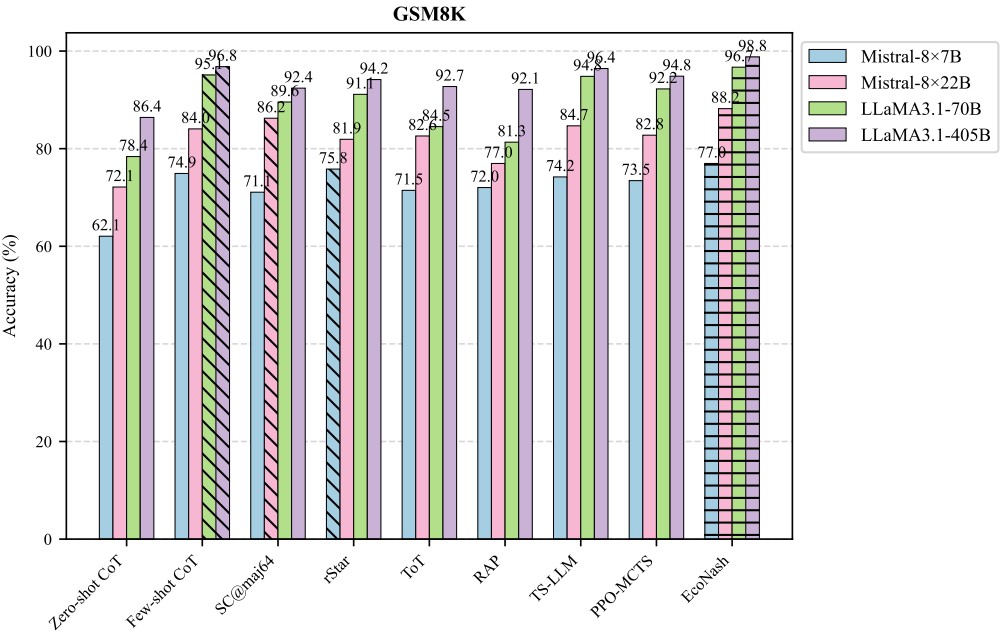

Figure 11: Accuracy Comparison of GSM8K

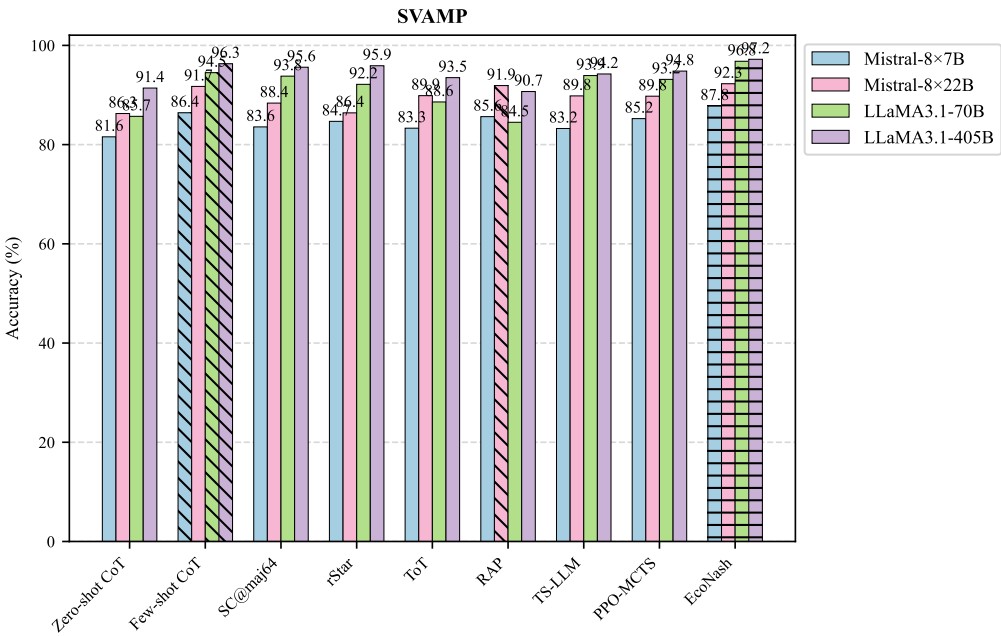

Figure 12: Accuracy Comparison of SVAMP

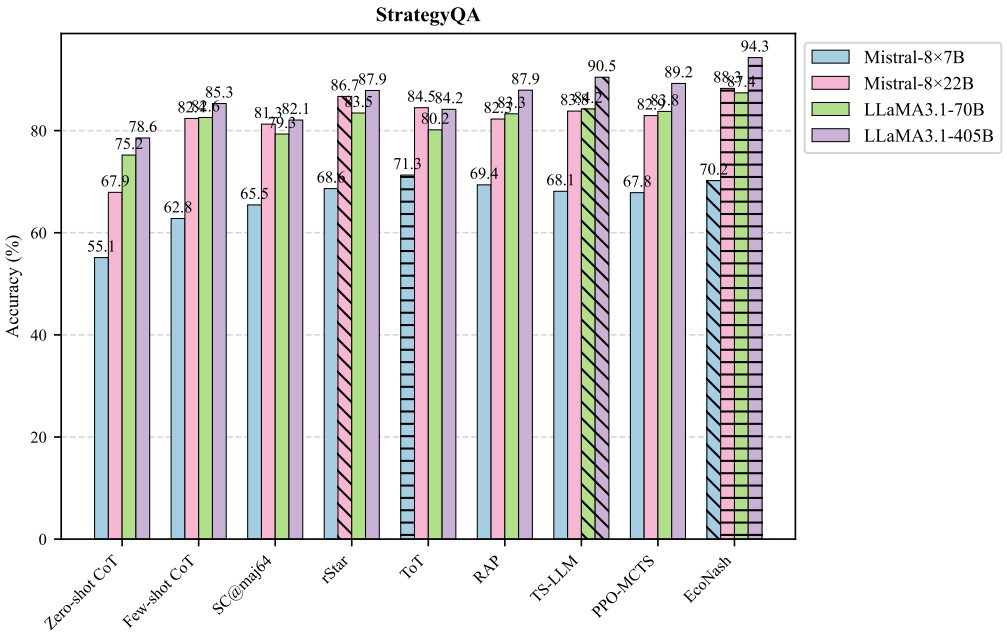

Figure 13: Accuracy Comparison of StrategyQA

# F. Example

## F.1. Case Study

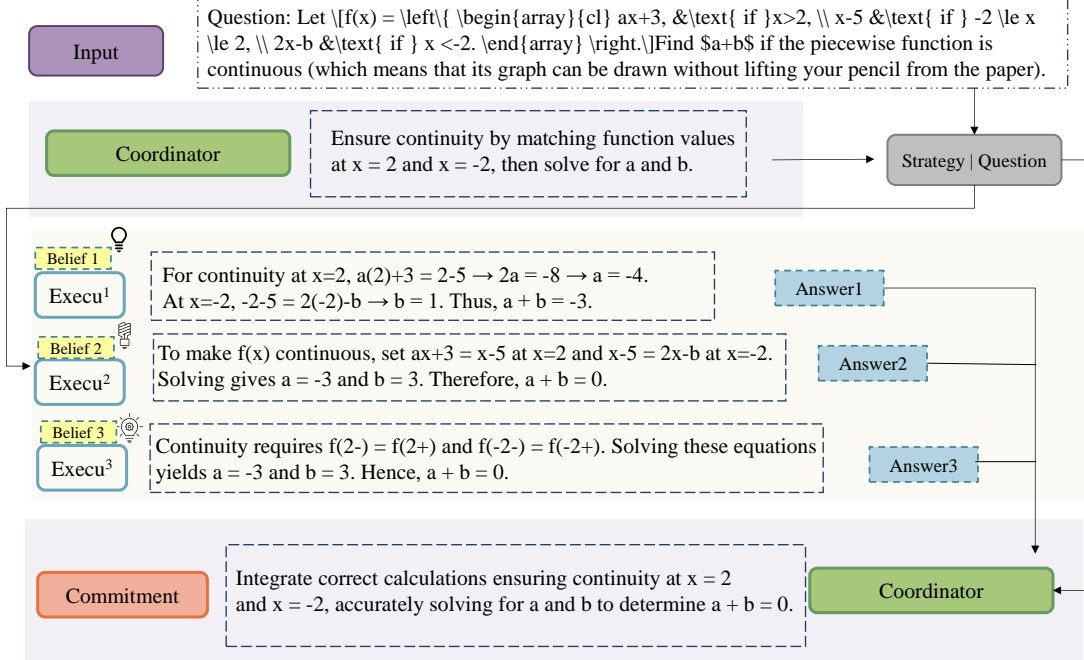

Figure 14: case study of math

## F.2. Strategy Examples

### F.2.1. GSM8K

---

**GSM8K Problem 1:**

**Q1:** John buys 3 pizzas for \$12 each. If he gives the delivery person a 20% tip on the total, how much did he spend in total?

**S1:** Calculate pizza subtotal first. Add 20% of subtotal for tip. Sum for final amount.

**F1:**
1. Pizza cost = \$? $\times$ ?
2. Tip = ? $\times$ subtotal
3. Total = subtotal + tip

$Strategy + Format : 35 tokens$

---

**GSM8K Problem 2:**

**Q2:** Janet saves twice as much money as Tom. If Tom saves \$45 per week, how much does Janet save in 5 weeks?

**S2:** Find Janet's weekly savings relative to Tom's. Multiply by number of weeks.

**F2:**
1. Janet weekly = ? $\times$ Tom
2. Total = weekly $\times$ weeks

$Strategy + Format : 28 tokens$

---

**GSM8K Problem 3:**

**Q3:** A factory produces 150 cars per day. If they increase production by 15% next month, how many cars will they produce in a 30-day month?

**S3:** Calculate production increase. Add to original. Multiply by days in month.

**F3:**
1. Increase = original $\times$ 15%
2. New daily = original + increase
3. Monthly = daily $\times$ days

$Strategy + Format : 36 tokens$

---

**GSM8K Problem 4:**

**Q4:** Alex has 240 marbles and gives $\frac{3}{8}$ of them to Sarah. Sarah then gives $\frac{1}{4}$ of her marbles to Tom. How many marbles does Sarah have left?

**S4:** Calculate Sarah's initial share. Find amount she gives to Tom. Subtract.

**F4:**
1. Sarah gets = total $\times \frac{3}{8}$
2. Sarah gives = her marbles $\times \frac{1}{4}$
3. Remaining = initial - given

$Strategy + Format : 39 tokens$

---

**GSM8K Problem 5:**

**Q5:** A train travels at 60 mph for 2.5 hours, then increases speed to 75 mph for 1.5 hours. What's the total distance traveled?

**S5:** Calculate distance for each speed separately using $d = r \times t$. Sum distances.

**F5:**
1. First distance = $speed_1 \times time_1$
2. Second distance = $speed_2 \times time_2$
3. Total = $d_1 + d_2$

$Strategy + Format : 36 tokens$

## F.2.2. MATH

**MATH Problem 1:**

**Q1:** In a bag of marbles, $\frac{3}{7}$ are blue and $\frac{2}{5}$ are red. The remaining 11 marbles are green. How many marbles are in the bag?

**S1:** Convert fractions to common denominator. Find the fraction for remaining color. Use given count to find total.

**F1:**
1. Convert to common denominator
2. Add converted fractions
3. Subtract from whole
4. Use remaining count to find total

$Strategy + Format : 32 tokens$

---

**MATH Problem 2:**

**Q2:** Find the area of a triangle with vertices at (0,0), (4,0), and (2,5).

**S2:** Use coordinate geometry method for area. Set up calculation matrix. Take final result.

**F2:**
1. Set up coordinate matrix
2. Calculate determinant
3. Apply area formula

$Strategy + Format : 28 tokens$

**MATH Problem 3:**

**Q3:** If $\log_2(x) = 3$ and $\log_2(y) = 4$, find $\log_2(xy)$.

**S3:** Apply logarithm properties. Combine given values. Express final result.

**F3:**
1. Write multiplication property
2. Substitute given values
3. Simplify result

$Strategy + Format : 26 tokens$

**MATH Problem 4:**

**Q4:** A circle has radius 6. Find the area of the sector formed by a $40°$ angle at the center.

**S4:** Convert angle measurement. Apply sector area formula. Simplify result.

**F4:**
1. Convert to radians
2. Write sector formula
3. Calculate final area

$Strategy + Format : 27 tokens$

**MATH Problem 5:**

**Q5:** Solve the equation: $2x^2 + 5x - 12 = 0$.

**S5:** Identify quadratic components. Apply standard formula. Solve for variables.

**F5:**
1. Identify coefficients
2. Setup quadratic formula
3. Calculate solutions

$Strategy + Format : 28 tokens$

## F.2.3. SVAMP

**SVAMP Problem 1:**

**Q1:** There are 56 books on the shelf. Tom puts 14 more books and Jane removes 22 books. How many books are on the shelf now?

**S1:** Track sequential changes. Apply additions and subtractions in order.

**F1:**
1. Add new books
2. Subtract removed books

$Strategy + Format : 25 tokens$

**SVAMP Problem 2:**

**Q2:** A box has 3 rows of chocolates. Each row has 4 chocolates. If 5 chocolates were eaten, how many are left?

**S2:** Calculate initial total. Subtract consumed amount.

**F2:**
1. Find total chocolates
2. Subtract eaten ones

$Strategy + Format : 23 tokens$

**SVAMP Problem 3:**

**Q3:** Mary has 5 times as many stickers as John. John has 12 stickers. How many stickers do they have together?

**S3:** Calculate second person's amount. Sum both quantities.

**F3:**
1. Find Mary's stickers
2. Add both totals

$Strategy + Format : 24 tokens$

**SVAMP Problem 4:**

**Q4:** A garden has 35 flowers. ($\frac{2}{7}$) are roses and ($\frac{3}{7}$) are tulips. How many flowers are neither roses nor tulips?

**S4:** Sum known fractions. Find remaining fraction. Calculate final count.

**F4:**
1. Add type fractions
2. Find remaining fraction
3. Calculate flower count

$Strategy + Format : 27 tokens$

**SVAMP Problem 5:**

**Q5:** Each child needs 3 pencils. If there are 23 children, how many boxes of 10 pencils should the teacher buy?

**S5:** Calculate total need. Convert to required units. Round appropriately.

**F5:**
1. Calculate total pencils
2. Divide by box size
3. Round to whole boxes

$Strategy + Format : 28 tokens$

**Note on Token Counts:**

- All problems follow consistent format: strategy + step-by-step format

- Strategy statements aim to be concise yet clear

- Format points provide framework without giving solutions

- Token ranges:
  - Shortest: 23 tokens (SVAMP Q2)
  - Longest: 39 tokens (GSM8K Q4)
  - Average: ($\sim$)30 tokens

