# OpenReview forum: "From Debate to Equilibrium: Belief‑Driven Multi‑Agent LLM Reasoning via Bayesian Nash Equilibrium"
_ICML.cc/2025/Conference — ICML 2025 poster_

### Official Review · Reviewer_t4Dv · 2025-03-13

**Overall Recommendation:** 3

**Summary:**

The paper introduces ECON, a hierarchical reinforcement learning framework designed to optimize multi-agent reasoning in Large Language Models (LLMs) by leveraging Bayesian Nash Equilibrium (BNE). ECON replaces inter-agent communication with a belief mechanism to save communication costs and make it easier to scale up the population of agents. LLMs integrate distributed reasoning and achieve centralized commitment by a Coordinator. Experiments show the proposed method surpasses the existing methods.

**Claims And Evidence:**

1. The paper claims that "they conceptually formalize BNE in Multi-LLM systems and instantiate it through a hierarchical optimization framework to improve collaborative reasoning", but the paper seems not to clearly describe how BNE is implemented in practice in the section 2.3 Framework of ECON.
2. One of the main claims of the paper is the scalability of the proposed method, while in Figure 4, the performance does not improve while increasing the number of Execution LLMs.

**Essential References Not Discussed:**

1. Belief mechanisms in MARL are not discussed, e.g. [1].
2. Multi-LLM-agent collaboration methods beyond debates are not included, e.g. [2][3].

[1] Wang, Yuanfei, Jing Xu, and Yizhou Wang. "ToM2C: Target-oriented Multi-agent Communication and Cooperation with Theory of Mind." International Conference on Learning Representations.
[2] Li, Guohao, et al. "Camel: Communicative agents for" mind" exploration of large language model society." Advances in Neural Information Processing Systems 36 (2023): 51991-52008.
[3] Guo, Xudong, et al. "Embodied LLM Agents Learn to Cooperate in Organized Teams." Language Gamification-NeurIPS 2024 Workshop.

**Experimental Designs Or Analyses:**

1. The Abaltion study does not analyze the effectiveness of different modules of the mix network. It may not be clear why the mix network should be designed in this way.
2. The paper mentions the method "outperforms single-LLM approaches by 10.9% and surpasses existing multi-LLM methods by 11.2%". Why the multi-LLM methods are even worse than single ones?

**Methods And Evaluation Criteria:**

1. I agree that inter-agent communication is one of the key concerns in LLM agents, but it does not mean it is better to remove such communication entirely. The belief mechanism proposed in the paper seems to follow the design in MARL, which lacks interpretability and generability. I believe the direction of this area should improve the efficiency of inter-agent communication instead of giving it up.
2. Benchmarks (GSM8K, MATH, TravelPlanner) are standard for reasoning tasks. Metrics (accuracy, token usage) align with the goals of efficiency and performance.

**Other Comments Or Suggestions:**

1. A space is missing on Page 1 " challenge(Liu et al., 2024b)."
2. On page 6 "Figure4"
3. On page 37 "Figure 14: case study of math"

**Other Strengths And Weaknesses:**

Strengths:
1. The paper explores a novel method to optimize the multi-agent reasoning with LLM.

Weaknesses:
1. The definition and effectiveness of belief in such tasks are not clear.
2. In Figure 2, the components of the framework are not clear enough. Which parts are based on LLM?

**Questions For Authors:**

Refer to the previous parts.

**Relation To Broader Scientific Literature:**

1. The paper incorporates the belief mechanism from MARL into LLM agents.
2. The paper further improves multi-agent debate method to executor-coordinator architecture.
3. The paper introduces Nash Equilibrium from game theory to LLM agents.

**Theoretical Claims:**

I did not find errors here.

---

> ### Author Rebuttal · Authors · 2025-04-01
>
> Dear Reviewer t4Dv, thanks for the feedback and suggestions, we will add clarification where needed as space permits.
>
> ### Q1:Explain the Implementation of BNE in Sec 2.3
>
> **Regarding the definition of belief**: belief represents each Execution LLM's policy derived from partial observations and local history, we use belief network to update belief. To enable coherent joint behavior, belief encoder aggregates individual belief states into group-level representation. Then the mix network coordinates this integrated belief information for global Q-value and compute loss and reward after commitment made. This architecture aligns with our formal definition of BNE in Sec 2.2, where optimizes agent policies through belief network updates until reaching a strategy profile where no agent can unilaterally improve its payoff given its beliefs.
>
> **Regarding the effectiveness** of belief-based coordination, we conduct additional experiments to response this concern. **The result of performance comparison with /without achieving BNE demonstrated an average performance improvement of 14%,** which can be found at the **response to reviewer 7mra Q1**.
> ### Q2: Explain the Result in Figure 4
>
> We would like to clarify that Figure 4 in Sec 3.4 intentionally demonstrates that **simply increasing the number of execution LLMs without appropriate coordination mechanisms can actually decrease performance.** We attribute this to the challenge faced by the Coordinator LLM in managing an excessive number of Execution LLMs in Sec 3.4.
>
> **To address this scalability challenge, we propose a global Nash equilibrium through local Nash equilibria by introducing additional coordinators in our manuscript (Sec 3.4).** This setup ensures that each Coordinator handles a reasonable amount of data, as demonstrated by the improvements reported in Figure 5. We provide the corresponding pseudocode and detailed explanation for scaling up ECON in Appendix A.4.
>
> ### Q3: Regarding The Inter-agent Communication in ECON
>
> > I believe the direction of this area should improve the efficiency of inter-agent communication instead of giving it up.
>
> We would like to clarify that ECON does not eliminate inter-agent communication entirely, but rather **adopts an incomplete-information perspective that minimizes communication** (as recognized by Reviewers sVjj and hmqe). ECON's optimization objective focuses on achieving consensus and constructing implicit communication among execution LLMs.
>
> **Although ECON is implicit inter-agent communication, it is not conflicted with explicit inter-agent communication**, we make additional experiment to demonstrate that by incorporating explicit interaction into ECON (which makes it become a complete information formation). The performance improve 1.1% while the token consumption increase 42.4% on average, demonstrate potential to achieve stronger performance with sufficient token budget.
> |Dataset|LLaMA3.1|Complete Info (%)|ECON (%)|Token Consumption|
> |---|---|---|---|---|
> |GSM8K|8B|81.4|80.3|+35.6%|
> ||70B|96.1|96.7|+42.7%|
> |GSM-Hard|8B|30.2|29.9|+62.3%|
> ||70B|53.6|51.4|+40.9%|
> |MATH|8B|59.6|60.4|+33.8%|
> ||70B|83.1|81.5|+39.4%|
> ### Q4: The Effectiveness of Different Modules of the Mix Network
>
> Theoretically, the design of our mixing network optimizes local policies to improve the global objective, monotonicity guarantee are demonstrated in Appendix A.5. **We make additional ablation experiments to validate this point by removing the concatenation of $e$ and remove belief encoder** to claim that each parts of ECON is essential, as shown in the experiments.The baseline result is ECON.
> |Dataset|LLaMA3.1|No concat(%)|No encoder(%)|
> |---|---|---|---|
> |GSM8K|8B|85.9(-1.8)|84.1(-3.4)|
> ||70B|93.6(-3.1)|91.1(-5.6)|
> |GSM-Hard|8B|24.9(-5.0)|21.7(-8.2)|
> ||70B|47.2(-4.2)|42.6(-8.8)|
> |MATH|8B|55.6(-4.8)|52.3(-7.1)|
> ||70B|77.0(-4.4)|75.3(-6.2)|
> ### Q5: Why are Multi-LLM Methods are Even Worse Than Single Ones?
>
> Our abstract needs clarification: **the 10.9% for MA-LLM is averaged across six datasets, while the 11.2% for Single is across five datasets (exclude TravelPlanner)**. Since single agent approach with open-source model have 0% pass rate on TravelPlanner, ECON's improvement ratio cannot be calculated. With TravelPlanner excluded, ECON's comparative results on five datasets are reported in Sec 3.2 which indicates the single-LLM methods are worse than multi-LLM methods.
>
> ### Q6: The Missing References Need to Discuss
>
> We agree that it's necessary to incorporate discussions of these works. We will update our related work section in the revised version.
>
> ### Q7: Clarification about Figure 2
>
> Figure 2 illustrates both the inference phase and optimization phase of ECON. The inference phase (left part) is primarily based on LLMs, where the Coordinator LLM and Execution LLMs work together to generate solutions. For a more detailed understanding of this process, we provide a comprehensive case study of the inference process in Appendix D.1.

---

### Official Review · Reviewer_hmqe · 2025-03-14

**Overall Recommendation:** 4

**Summary:**

This paper proposes a Multi-agent LLM framework (ECON) to improve the communication efficiency and consensus. It formulates multi-agent LLM as Decentralized Partially Observable Markov Decision Process. It reduces the token consumption from incomplete-information perspective, and optimizes towards Bayesian Nash Equilibrium to improve the consensus. The proposed method has a lower regret bound, making it possible to scale up effectively. Experimental results on six reasoning tasks show that ECON surpasses single-agent solutions and outperforms existing multi-agent approaches with a lower token consumption.

**Claims And Evidence:**

This paper claims the proposed ECON can ensure the Multi-agent LLM system can converge towards Bayesian Nash Equilibrium, thus enhancing the degree of consensus and performance.

**Essential References Not Discussed:**

N/A

**Experimental Designs Or Analyses:**

The experimental section can be divided into 4 parts:
(1) performance against baseline models across 5 reasoning tasks with 3 llama models and 1 mistral model, and complex planning on Travelplanner with GPT-4 turbo
(2) strong coordinator with weak execution & weak coordinator with strong execution to evaluate the heterogeneous Execution
(3) the result of scale-up agent numbers.
(4) ablation study.

These experiments and analyses are comprehensive and reasonable, demonstrating the effectiveness of ECON in different settings.

**Methods And Evaluation Criteria:**

ECON uses a Coordinator-Executor architecture. Each Execution LLM keeps a brief network that updates its belief state with the local trajectory and the current observation. A shared belief encoder aggregates the belief states from all agents to model coherent joint behavior.

However, Section 2 mainly focuses on the optimization phase and it is not clear how the inference phase works in ECON. What is the observation of each agent, what is the strategy provided by the coordinator, how Coordinator aggregates the answers, etc are not mentioned. This make the method part confusing.

The evaluation metric is accuracy on four mathematical reasoning datasets and the common sense reasoning dataset. On TravelPlanner, the metric is the final pass rates.

**Other Comments Or Suggestions:**

Figure 2, there is no red gradient flow.
Figure 3, it is better to group the bins by LLMs instead of the baseline methods because it should highlight the difference between different baselines rather than different LLMs.

**Other Strengths And Weaknesses:**

Strengths:
The idea is interesting and novel.
Theoretical proofs are provided to guarantee the effectiveness of ECON.
The experiments are comprehensive and the findings are insightful.

Weaknesses: While Section 2 provides the formulation and optimization of ECON, it is not clear how ECON works during the inference phase. The meaning of notations such as O_i and e_i and their instantiation in a concrete task should be introduced (e.g. text or hidden representation)

**Questions For Authors:**

the prompt embedding e_i = [T_i, p_i], where T_i is the temperature and p_i is the threshold for repetition. How do T_i and p_i construct the token embedding for the prompt?
How does ECON scale up to more agents? Does it require additional training?

**Relation To Broader Scientific Literature:**

It is related to the multi-agent reinforcement learning and methods in game theory

**Theoretical Claims:**

I didn't check the correctness of the correctness of the proof

---

> ### Author Rebuttal · Authors · 2025-04-01
>
> Dear Reviewer hmqe:
> Thanks for your constructive review, we provide some response regarding your question:
>
> ### Q1: About the inference phase of ECON
>
> > Section 2 mainly focuses on the optimization phase and it is not clear **how the inference phase works in ECON**. What is the observation of each agent, what is the strategy provided by the coordinator, how Coordinator aggregates the answers, etc are not mentioned.
>
> We provide a more intuitive and detailed explanation of our framework below:
>
> 1. Intuitive Framework Explanation
>
> During the **inference phase** of ECON, a Coordinator LLM generates an informative strategy and a format based on the input question. These are then disseminated to the Execution LLMs, which independently produce their respective answers. Finally, the Coordinator LLM aggregates these answers to form a final commitment. A detailed case study demonstrating this inference process is provided in Appendix D.2.
>
> In the **optimization phase** of the ECON framework, we update each Execution LLM's belief state through its belief network, aggregate these belief states via the belief encoder to form group-level information, then the mixing network outputs the final Q_tot. After the current answer is completed, we calculate loss and reward based on the commitment, which are then used to update the parameters of each Execution LLM's belief network and the belief encoder. We describe our specific DEC-POMDP components setting as follows:
>
> 1. Detailed DEC-POMDP Components
>
> **Partial Observation:**
> In our framework, each agent receives **only three types of information: the question it needs to answer, the format and strategy provided by the coordinator, and the final commitment** (aggregated response) after the coordinator combines all LLM answers. This setting prevents Execution LLMs from directly accessing the outputs of others.
>
> **Local History:**
> Local history includes its actions and received observations. This local history enables the agent to learn from past behaviors and environmental feedback, refining its situational understanding and updating its belief state accordingly.
>
> **Action: Prompt embedding**
> The **action space** is represented by the **prompt embedding**, which encodes the agent's action output. The prompt embeddings directly influence the LLM's generation process, adjusting the balance between exploration and exploitation.
>
> **State Transition:**
> The state transition in our framework is essentially the update of the agent's **belief state**. The belief network updates the belief state based on the agent's local history and current observation, influencing future actions.
>
> ### Q2: Clarification about prompt embedding:
>
> >the prompt embedding e_i = [T_i, p_i], where T_i is the temperature and p_i is the threshold for repetition. How do T_i and p_i construct the token embedding for the prompt?
>
> Thank you for your insightful comments. We agree that our description of the prompt embedding requires clarification.
>
> - The prompt Embedding actually the action generated by belief state, we use this expression because it embeds control parameters that directly influence the Execution LLM's generation process. We will rename this in the revised manuscript to avoid confusion about embedding of answer output.
>
> ### Q3: About the scalability of ECON:
>
> > How does ECON scale up to more agents? Does it require additional training?
> We would like to clarify how we make our framework scalable and yes, scalable ECON require additional training:
>
> - Our solution enhances scalability by forming a **global Nash equilibrium through local Nash equilibria** by introducing additional coordinators. Simply increasing the number of Execution LLMs would cause performance degradation since coordinator LLMs cannot handle excessive information (especially for weaker models).
>  - As shown in **Figure 4**, merely increasing Execution LLMs causes performance decline as coordinator LLMs struggle to manage numerous agents, particularly with weaker models.
> - Results in **Figure 5** demonstrate that our scaled-up system (9 Execution LLMs, 3 coordinators, 1 central LLM) achieved an **18.1% improvement** over the baseline system (3 Execution LLMs, 1 coordinator), indicating potential for further scaling.
> - More details about ECON's scalability can be found in **Section 3.4**, with detailed explanation and pseudocode in **Appendix A.4**.
>
> ## Q4: Figure Presentation Issues
>
> > In Figure 2, the red gradient flow is missing. In Figure 3, it would be more effective to group the bins by LLMs rather than baseline methods, as this would better highlight differences between baselines rather than between different LLMs.
>
> **Response:**
> We appreciate your constructive feedback. We will address these visualization issues in the revised manuscript by adding the missing red gradient flow to Figure 2 and reorganizing Figure 3 to group bins by LLMs, which will indeed better emphasize the comparative performance.

---

> > ### Comment · Reviewer_hmqe · 2025-04-03
> >
> > Thanks for the authors' detailed rebuttal, which has solved most of my concerns.
> >
> > In total, I like the idea of this work but will suggest rewriting the method section to include these necessary descriptions and make it easier to understand.  Besides, the additional training for scaling up makes the proposed method less flexible and adaptive, which is one of the main limitations of ECON. Discussion about the training cost for the convergence in terms of different numbers of agents can provide more insights into this aspect.

---

> > > ### Author Response · Authors · 2025-04-04
> > >
> > > Dear Reviewer hmqe,
> > >
> > > We sincerely appreciate your thoughtful review and positive assessment of our work. In response to your valuable feedback, we will restructure and revise the method section (Section 3.1) to provide clearer definitions and more comprehensive descriptions. The updated section will follow a more logical flow, enhancing the accessibility and readability of the final manuscript.
> > >
> > > Regarding the discussion about the training cost for convergence in terms of different numbers of agents, we have conducted an additional comparison between ECON (1 coordinator, 3 execution LLMs) and scaled ECON (1 central model, 3 coordinators, 9 execution LLMs) using LLaMA 3.1 70B on the MATH dataset. The scaling increases the number of trainable parameters from 1.7M to 8.9M (5.2×), requiring 1.7× more convergence episodes while achieving an 18.1% overall performance improvement. We will expand our conclusion section to include a more thorough discussion of this trade-off by providing detailed analysis on the correlation between training costs and the number of agents, as well as examining their convergence patterns under different scaling configurations.
> > >
> > > Thank you once again for your constructive feedback, which has been invaluable in improving the quality and impact of our work.
> > >
> > >
> > > | Parameter | ECON | Scaled ECON | Change |
> > > | --- | --- | --- | --- |
> > > | **Training Configuration** |  |  |  |
> > > | Episodes  | 150 | 250 | ×1.67 |
> > > | Buffer Size | 32 | 64 | ×2.0 |
> > > | Batch Size | 16 | 24 | ×1.5 |
> > > | Update Interval | 8 | 12 | ×1.5 |
> > > | **Network Architecture** |  |  |  |
> > > | Entity Dimension | 256 | 384 | ×1.5 |
> > > | Belief State Dimension | 128 | 192 | ×1.5 |
> > > | Attention Heads | 4 | 8 | ×2.0 |
> > > | Transformer Blocks | 2 | 3 | ×1.5 |
> > > | Feed-forward Size | 1024 | 2048 | ×2.0 |
> > > | **Model Complexity** |  |  |  |
> > > | Trainable Parameters | 1.7M | 8.9M | ×5.2 |
> > > | Convergence Episodes| 99| 164| ×1.7 |

---

### Official Review · Reviewer_sVjj · 2025-03-14

**Overall Recommendation:** 3

**Summary:**

The paper introduces ECON, a hierarchical reinforcement learning framework that optimizes multi-agent reasoning in Large Language Models (LLMs) by leveraging Bayesian Nash Equilibrium (BNE) under incomplete information. By modeling collaboration as a Decentralized Partially Observable Markov Decision Process (DEC-POMDP), ECON ensures each LLM agent independently generates solutions based on local beliefs and a shared coordinator strategy, minimizing inter-agent communication. The framework employs a Coordinator-Executor architecture: Execution LLMs produce answers using belief networks that model probabilistic expectations of others’ behaviors, while a Coordinator LLM aggregates responses into a global commitment. Theoretical contributions include proving BNE existence and achieving a sublinear regret bound, outperforming linear regret in non-BNE methods. Key innovations include belief networks for reducing token overhead, dynamic reward mechanisms balancing task performance and collaboration, and hierarchical Nash coordination for scalability.

**Claims And Evidence:**

**Sublinear Regret**

The existence of Bayesian Nash Equilibrium (BNE) is rigorously proven using Glicksberg’s Fixed Point Theorem (Appendix A.1), with assumptions like strategy space compactness and quasi-concave payoffs explicitly stated. The sublinear regret bound is derived under standard RL assumptions (bounded rewards, concentrability), supported by a detailed decomposition of Q-value differences (Appendix B.2).

Critical assumptions like concentrability (Assumption A.8) and posterior alignment (Assumption A.4) are central to the regret analysis but not empirically validated.

**Empirical Results**

Performance gains over single-LLM (10.9%) and multi-agent baselines (11.2%) are validated across six benchmarks (Tables 1–3, Figures 3–5), with detailed task setups (Appendix B.5) and hyperparameters (Appendix B.6). Token efficiency (21.4% reduction vs. Multi-Agent Debate) is demonstrated via token usage tables (Table 3), though prompts and strategies are standardized (Appendix D).

While performance improvements are reported, statistical significance tests (e.g., confidence intervals, p-values) are absent.

**Essential References Not Discussed:**

No.

**Experimental Designs Or Analyses:**

I have checked the its main experiments (Sec 3.2), weaker or stronger execution llms (Sec 3.3), ans scaling up to multiple execution llms by hierarchical coordination (Sec 3.4). and ablation study on different compoent of the model (Sec 3.5). I think the results are promising.

**Methods And Evaluation Criteria:**

**Benchmarks**:

Tasks like GSM8K, MATH, and TravelPlanner are standard for evaluating reasoning and planning in LLMs, ensuring comparability with prior work.Including heterogeneous model experiments (Table 2) tests robustness, though deeper analysis of weaker models’ contributions would strengthen validity.

**Metrics**:

Accuracy and token consumption directly measure performance and efficiency, addressing the paper’s core claims. Scalability tests (up to 9 LLMs) validate the framework’s practical utility for large ensembles.

**Other Comments Or Suggestions:**

In the Appendix, the section numbering is incorrect. For example, "Together API Integration for ECON" should be section C I think.

**Other Strengths And Weaknesses:**

No.

**Questions For Authors:**

- Does the results of ECON in Table.3 contains the token usage of coordinator models?
- Table 4 seems to be confusing for me, why R3 does not appear in the column name, and where is S3 in the table?
- I suggest the author to explain what the local history $\tau$ and partial observation $O$ consist of.
- What is U in the definition of Sec 2.2, can you write the math form? What $\mathcal{R}$ consists of? I think $\mathcal{R}=\{ r_{coordinator}, r_1,\cdots, r_n \} $? I also notice that the notation $R$ is used in both the reward design and regret $R(T)$, please avoid reusing notation names.
- is $\theta$ in Sec 2.1 related to $\theta^B$ in Sec 2.2? It seems that $\theta^B$ is model's parameters but $\theta$ is something like the concatenation of $b_i$ and $\tau_i$. Can you write the math definition of $\theta$? I think it should be $(\tau_i, O_i)$, i.e., past history and current obervation.
- The equation in Sec 2.3 is a little bit confusing. $B_i$ output $Q_i$ and $e_i$.
    - $e_i$ be called a prompt embedding? Based on your writing, I think $T_i$ and $p_i$ are scalars and $e_i$ is a two-dimensional vector, I wonder why this can be called "embedding", and can you explain why you concatenate it with ${\bf E}_i$. As stated in the paper, $T_i$ and $P_i$ control the sampling, so why do you connect it with belief encoding?
    - $Q_i$ is the output of $B_i$, therefore, $\phi$ should be part of $\theta^B$, I think? But the current writing does not relfect this relation and it is confusing where $\phi$ comes from.

**Relation To Broader Scientific Literature:**

- Introduces Bayesian Nash Equilibrium (BNE) to formalize consensus in multi-agent LLMs,

- Proposes a Coordinator-Executor architecture where a Coordinator LLM guides Execution LLMs via strategies, combining CTDE’s centralized coordination with the flexibility of LLM-based reasoning. This extends hierarchical RL to static, non-trainable LLM agents.

- Bridges game-theoretic equilibria with modern LLM ensembles, offering a blueprint for principled multi-agent reasoning.

**Theoretical Claims:**

I have checked the proof, but I think the proof is hard to understand. For example, in Appendix B.2, the proof is an outline rather than an rigorous proof. I think this part should be rewritten because it is one of the key theoretical contributions: sublinear convergence rate.

---

> ### Author Rebuttal · Authors · 2025-04-01
>
> Dear reviewer sVjj, we'd like to thank you for your careful readings and valuable comments, we provide point to point response as follow:
>
> ### Q1: The proof of sublinear convergence rate (Appendix B.2)
>
> We acknowledge that the proof in Appendix B.2 would benefit from a more rigorous presentation. In the revised manuscript, we enhance this proof with the following improvements, while we cannot include the rewritten proof in this rebuttal due to space limitations and URL limitation, we provide the procedure as follow:
>
> - Formally justify the $O(t^{-1/2})$ convergence rates for both Q-function estimation errors and policy suboptimality through stochastic approximation theory and convex optimization results.
> - Explicitly demonstrate how our learning rate schedule $\eta_t = \eta_0/\sqrt{t}$ ensures these convergence properties under stated assumptions.
> - Include the complete mathematical derivation from regret definition to final bound, showing:
>     - Precise Q-value decomposition into error terms
>     - Rigorous bounding of each error component
>     - Explicit summation across agents and time steps
>     - Formal harmonic sum bound: $\sum_{t=1}^T 1/t^{1/2} \leq 2\sqrt{T}$
> - We present the complete derivation of the final $O(N\sqrt{T}/(1-\gamma))$ bound with all intermediate mathematical steps included.
>
> ### Q2: The token consumption of ECON
>
> Yes, the token usage reported in Table 3 including the coordinator LLM's output strategy and formatting instructions, the execution LLMs' answers, and the coordinator LLM's aggregated answers. We provide a case study about ECON Inference process in Appendix D.1 and a token usage for strategy formulation and formatting in Appendix D.2.
>
> ### Q3: Regarding the confusion caused by Table 4
>
> We apologize for the presentation of Table 4. You are correct that R3 and S3 are missing from the table, which makes it difficult to interpret. We have provided a revised Table 4 as follow:
> |$R_1$|$R_2$|$R_3$|ECON|
> |-|-|-|-|
> |✓|✗|✓|77.55|
> |✓|✗|✗|74.32|
> |✓|✓|✗|76.21|
> |Random|||62.71|
>
> |$S_1$|$S_2$|$S_3$|ECON|
> |-|-|-|-|
> |✓|✗|✗|71.35|
> |✗|✓|✗|72.31|
> |✗|✗|✓|81.47|
> ### Q4: Explanation of notations
>
> > I suggest the author to explain what the local history and partial observation consist of.
> >
>
> We make a more detailed explanation about local history and partial observation, as the limitation of space, **please refer to the response of reviewer 7mra Q2**.
>
> > What is U in the definition of Sec 2.2, can you write the math form? What R consists of?
> >
>
> **Utility Function**: In our framework, the utility function $U_i$ represents the expected cumulative reward for agent i, calculated as:
>
> $$U_i(\pi_i(\theta_i), \pi_{-i}(\theta_{-i}), \theta_i, \theta_{-i}) = \mathbb{E} \left[\sum_{t=0}^{\infty} \gamma^t r_i^t \mid \pi_i, \pi_{-i}, \theta_i, \theta_{-i} \right],$$
>
> where $r_i^t$ is the reward at time step $t$, the components of this reward function are detailed in Sec 2.3 under "Reward Design." And $R$ consists of $[r_i]$, i.e., $r_1, r_2, ..., r_n$.
>
> > Can you write the math definition of θ?
> >
>
> Yes you are right, the θ in Sec 2.1 defined as  local history $\mathbf{\tau}_i^t$ and observation $O_i^t$.
>
> ### Q5: Clarification of Sec 2.3
>
> > I wonder why this can be called "embedding", and can you explain why you concatenate it with Ei
> It is confusing where ϕ comes from.
> >
>
> #### **Regarding Prompt Embeddings**
>
> - The prompt Embedding actually the action generated by belief state, we use this expression because it embeds control parameters that directly influence the LLM's generation process. We will rename this in the revised manuscript to avoid confusion about embedding of answer output.
> - Our approach encodes the action $e$ and the global state $E_t$ to jointly optimize the global $Q$-function $Q_{tot}$. The former focuses on local action decisions, enhancing diversity and aiding cooperative exploration, while the latter captures comprehensive group context to foster more efficient inter-agent coordination. We make additional ablation experiments to validate this point by removing the concatenation of $e$ and remove belief encoder; **please refer to the response of reviewer t4DV Q4**.
>
> #### **Regarding φ**: You are correct that there is an unclear relationship between $\phi$ and $\theta_i^B$. In our revised manuscript, we will explicitly denote $\phi_i \subset \theta_i^B$ to indicate that Q-value function parameters are a subset of the belief network parameters.
>
> ### Q6: Deeper analysis of weaker models
>
> > Though deeper analysis of weaker models' contributions would strengthen validity.
>
> We have conducted additional experiments with smaller language models and found that our framework still provides significant improvements over baseline approaches, the comparison is made with few-shot CoT.
> |Dataset|Model|Few-shot CoT (%)|ECON (%)|
> |---|---|---|---|
> |GSM8K|QWEN2.5 3B|79.1|84.9|
> ||LLaMA3.1 8B|84.5|87.7|
> |GSM-Hard|QWEN2.5 3B|19.7|21.3|
> ||LLaMA3.1 8B|27.6|29.9|
> |MATH|QWEN2.5 3B|42.6|49.7|
> ||LLaMA3.1 8B|51.9|60.4|

---

### Official Review · Reviewer_7mra · 2025-03-16

**Overall Recommendation:** 3

**Summary:**

This paper introduces ECON, a multi-agent framework designed to enhance the reasoning capabilities of LLMs. ECON models the multi-LLM setup as a DEC-POMDP with incomplete information, employing a Bayesian Nash Equilibrium perspective. Specifically, multiple “Execution LLMs” reason in parallel, each maintaining its own belief network and generating local solutions under partial information. A separate “Coordinator LLM” orchestrates consensus by aggregating and evaluating these local solutions, issuing a guidance to all agents. This structure aims to achieve a BNE, in which no single LLM agent can unilaterally improve its outcome, given its beliefs about the other agents.

**Claims And Evidence:**

Yes. In general, the claims are supported by (1) regret bounds derived from RL theory and DEC-POMDP frameworks, and (2) empirical results across diverse tasks.

**Essential References Not Discussed:**

This work did a good job in reviewing and discussing related references.

**Experimental Designs Or Analyses:**

The authors test on well-established datasets, including math reasoning, commonsense QA set, and planning, using models with different sizes.

**Methods And Evaluation Criteria:**

Yes. The hierarchical reinforcement learning approach and the evaluation (both theoretical and empirical) are

**Other Comments Or Suggestions:**

NA

**Other Strengths And Weaknesses:**

I have a mixed feeling about this work, stated in the following:

Strength:
- I strongly believe that incorporating a more principled approach into multi-agent LLM collaboration (e.g., a game-theoretical one) is highly valuable. This work makes advances in this direction and can inspire future works.

- This work is a nice combination of theory and practice. In particular, both theoretical regret analyses and empirical test results are provided.

Weakness:
- The presentation can be largely improved, in particular, providing more clarity and intuition on the framework. For example, the mapping into DEC-POMDP is a key step, which is not presented in a very abstract way without explanation of related concepts (e.g., how is the partial observation received, what is the state transition, why is the action "generating prompt embeddings"). With the lack of these, the understanding of the connection is very hard.

- Also, the learning target should also be explained better. In general, we wish to have the system perform well in an aggregated fashion (i.e., we only care about the final output from the aggregator), while this work focus on BNE. A better connection should be drawn.

- The scalability of the proposed training method should also be discussed. The previous works, while less principled, can be flexibly extended without training. This work, however, requires certain training steps to enable collaboration. It is unclear whether such training is valuable in terms of flexibility and efficiency.

**Questions For Authors:**

My name concerns are listed in "Other Strengths And Weaknesses". It would be much appreciated if the authors can discuss these aspects.

**Relation To Broader Scientific Literature:**

In general, this extends prior “multi-agent debate” by proposing incomplete-information modeling and rigorous regret analysis, bridging a gap between purely heuristic debate approaches and formal MARL frameworks.

**Theoretical Claims:**

I have skimmed through the proof of the theoretical results and believe they are intuitively correct.

---

> ### Author Rebuttal · Authors · 2025-04-01
>
> Dear Reviewer 7mra:
> Thanks for your constructive review, we provide point to point response regarding your question:
>
> ### Q1: The Clear Definition of DEC-POMDP and the ECON Framework
>
> **Reply:** We provide a more intuitive and detailed explanation of our framework below:
>
> ----
>
> ### Detailed DEC-POMDP Component
>
> **Partial Observation:**
> In our framework, each agent receives **only three types of information: the question it needs to answer, the format and strategy provided by the coordinator, and the final commitment** (aggregated response) after the coordinator combines all LLM answers. This setting prevents Execution LLMs from directly accessing the outputs of others.
>
> **Local History:**
> Local history includes its actions and received observations. This local history enables the agent to learn from past behaviors and environmental feedback, refining its situational understanding and updating its belief state accordingly.
>
> **Action: Prompt embedding**
> The **action space** is represented by the **prompt embedding**, which encodes the agent's action output. The prompt embeddings directly influence the LLM's generation process, adjusting the balance between exploration and exploitation.
>
> **State Transition:**
> The state transition in our framework is essentially the update of the agent's **belief state**. The belief network updates the belief state based on the agent's local history and current observation, influencing future actions.
>
> ----
> ### Intuitive Framework Explanation
>
> During the **inference phase** of ECON, a Coordinator LLM generates an informative strategy and a format based on the input question. These are then disseminated to the Execution LLMs, which independently produce their respective answers. Finally, the Coordinator LLM aggregates these answers to form a final commitment. A detailed case study demonstrating this inference process is provided in Appendix D.1.
>
> In the **optimization phase** of the ECON framework, we update each Execution LLM's belief state through its belief network, aggregate these belief states via the belief encoder to form group-level information, then the mixing network outputs the final Q value. After the current answer is completed, we calculate loss and reward based on the commitment, which are then used to update the parameters of each Execution LLM's belief network and the belief encoder.
>
> ---
>
> ### Q2: Explanation of the Learning Target of ECON Framework
>
> **Reply:** We clarify how setting BNE as the learning target leads to better aggregated answers as follows:
>
> As mentioned in MAD, the key improvement in Multi-Agent Debate lies in **agreement intensity,** indicates the degree to which agents agree with each other can provide significant performance gains. This principle underlies our approach, where **we set BNE as the optimization target to establish consensus among the agents.** We then analyze the total Bayesian regret of the joint policy (i.e., the aggregated answer output by the MA-LLM system) based on learning towards BNE in **Lemma 2.2** and **Appendices B.2–B.3**.
>
> To further validate that learning towards BNE can lead to better aggregated answers, we provide additional experiments showing the actual performance differences of the ECON framework before and after achieving BNE as follow:
>
>
> | Dataset | LLaMA3.1| Without BNE (%) | With BNE (%) |
> |-|-|-|-|
> | GSM8K |8B | 74.38 | 80.33 |
> | |70B | 82.12 | 96.61 |
> | | 405B | 92.36 | 99.17 |
> | GSM-Hard | 8B | 21.73 | 30.71 |
> | | 70B | 43.58 | 60.26 |
> | | 405B | 51.54 | 65.91 |
> | MATH | 8B | 55.92 | 71.45 |
> | | 70B | 74.47 | 87.31 |
> | | 405B | 82.31 | 94.78 |
>
> ----------
>
> ### Q3: Discuss the Issue of ECON Scalability
>
> **Reply:**  We would like to clarify how we make our framework scalable:
>
> - Our solution enhances scalability by forming a **global Nash equilibrium through local Nash equilibria** by introducing additional coordinators. Simply increasing the number of Execution LLMs would cause performance degradation since coordinator LLMs cannot handle excessive information (especially for weaker models).
> - As shown in **Figure 4**, merely increasing Execution LLMs causes performance decline as coordinator LLMs struggle to manage numerous agents, particularly with weaker models.
> - Results in **Figure 5** demonstrate that our scaled-up system (9 Execution LLMs, 3 coordinators, 1 central LLM) achieved an **18.1% improvement** over the baseline system (3 Execution LLMs, 1 coordinator), indicating potential for further scaling.
> - More details about ECON's scalability can be found in **Section 3.4**, with detailed explanation and pseudocode in **Appendix A.4**.

---

### Decision · Program_Chairs · 2025-05-01

**Decision:**

Accept (poster)

**Comment:**

After discussion with the authors and updating their scores, reviewers unanimously agree on accepting this paper, with final scores of [3,3,4,3].

Strengths identified by the reviewers include:
- Theoretical results
  - “extends prior “multi-agent debate” by proposing incomplete-information modeling and rigorous regret analysis, bridging a gap between purely heuristic debate approaches and formal MARL frameworks.” (7mra)
  - “The existence of Bayesian Nash Equilibrium (BNE) is rigorously proven using Glicksberg’s Fixed Point Theorem (Appendix A.1), with assumptions like strategy space compactness and quasi-concave payoffs explicitly stated. The sublinear regret bound is derived under standard RL assumptions (bounded rewards, concentrability), supported by a detailed decomposition of Q-value differences (Appendix B.2).” (sVjj)
  - “The proposed method has a lower regret bound, making it possible to scale up effectively” (hmqe)
- Empirical results
  - experiments are “comprehensive and reasonable” (hmqe)
  - “The authors test on well-established datasets” like GSM8K, MATH, and TravelPlanner, enabling comparison with prior work (sVjj, 7mra)
  - “Performance gains over single-LLM (10.9%) and multi-agent baselines (11.2%) are validated across six benchmarks (Tables 1–3, Figures 3–5), with detailed task setups (Appendix B.5) and hyperparameters (Appendix B.6). Token efficiency (21.4% reduction vs. Multi-Agent Debate) is demonstrated via token usage tables (Table 3), though prompts and strategies are standardized (Appendix D).” (sVjj)
- “This work did a good job in reviewing and discussing related references” (7mra)
- “The idea is interesting and novel.” (hmqe)
- “the findings are insightful.” (hmqe)

Many of the weaknesses identified by the reviewers related to the clarity of the paper:
- “the paper seems not to clearly describe how BNE is implemented in practice in the section 2.3 Framework of ECON.” (t4Dv)
- “I have checked the proof, but I think the proof is hard to understand. For example, in Appendix B.2, the proof is an outline rather than an rigorous proof. I think this part should be rewritten because it is one of the key theoretical contributions: sublinear convergence rate”. (sVjj)

In the rebuttal, the authors provided clarifications to the reviewer’s questions, and spelled out a high-level strategy for a revised proof to answer reviewer sVjj’s concern. Several errors (e.g. with Table 4) were corrected. They also provided several additional experiments with different ablations, using smaller models with CoT, scaling up to achieve even better performance improvements, etc.

Given this evidence, I am recommending accepting the paper, but would strongly encourage the authors to revise the paper to clarify the mistakes and issues that were identified in the review process.